# What Are Good Positional Encodings for Directed Graphs?

**Yinan Huang**
Georgia Institute of Technology
`yhuang903@gatech.edu`

**Haoyu Wang**
Georgia Institute of Technology
`haoyu.wang@gatech.edu`

**Pan Li**
Georgia Institute of Technology
`panli@gatech.edu`

## Abstract

Positional encodings (PEs) are essential for building powerful and expressive graph neural networks and graph transformers, as they effectively capture the relative spatial relationships between nodes. Although extensive research has been devoted to PEs in undirected graphs, PEs for directed graphs remain relatively unexplored. This work seeks to address this gap. We first introduce the notion of *Walk Profile*, a generalization of walk-counting sequences for directed graphs. A walk profile encompasses numerous structural features crucial for directed graph-relevant applications, such as program analysis and circuit performance prediction. We identify the limitations of existing PE methods in representing walk profiles and propose a novel *Multi-q Magnetic Laplacian PE*, which extends the Magnetic Laplacian eigenvector-based PE by incorporating multiple potential factors. The new PE can provably express walk profiles. Furthermore, we generalize prior basis-invariant neural networks to enable the stable use of the new PE in the complex domain. Our numerical experiments validate the expressiveness of the proposed PEs and demonstrate their effectiveness in solving sorting network satisfiability and performing well on general circuit benchmarks. Our code is available at `https://github.com/Graph-COM/Multi-q-Maglap`.

## 1 Introduction

Positional encoding (PE) (Vaswani et al., 2017; Su et al., 2024), which refers to the vectorized representation of token positions within a series of tokens, has been widely integrated into modern deep learning models across various data modalities, such as language modeling (Vaswani et al., 2017), vision tasks (Dosovitskiy et al., 2020), and graph learning (Dwivedi et al., 2023; Rampášek et al., 2022). The key advantage of PE is its ability to preserve crucial positional information of tokens, thereby complementing many downstream position-agnostic models like transformers (Yun et al., 2019) and graph neural networks (GNNs) (Li & Leskovec, 2022; Lim et al., 2022). For regularly ordered data, such as sequences or images, defining PE is relatively straightforward; for instance, one can use sinusoidal functions of varying frequencies (known as Fourier features) as PE (Vaswani et al., 2017). In contrast, designing PE for structured graph data is more challenging due to the lack of canonical node ordering in graphs.

Designing effective positional encodings for graphs (Dwivedi et al., 2023; Kreuzer et al., 2021; Beaini et al., 2021; Lim et al., 2022; Dwivedi et al., 2021; Rampášek et al., 2022; Li et al., 2020; Wang et al., 2022) is significant because graph PEs can be used for building powerful graph transformers as well as improving the expressive power of GNNs (Li et al., 2020; Wang et al., 2022; Lim et al., 2022). In particular, for undirected graphs, Laplacian positional encoding (Lap-PE) (Dwivedi et al., 2023; Kreuzer et al., 2021), derived from the eigenvalues and eigenvectors of graph Laplacians, is adopted by many state-of-the-art graph machine learning models (Rampášek et al., 2022; Chen et al., 2022; Lim et al., 2022; Huang et al., 2023). Lap-PE is powerful as it well preserves the information of the graph structure and can express various distance features defined on undirected graphs: examples such

as diffusion distance (Coifman & Lafon, 2004; Bronstein et al., 2017; Wang et al., 2024), resistance distance (Xiao & Gutman, 2003) and biharmonic distance (Lipman et al., 2010; Kreuzer et al., 2021) are crucial structural features widely used in many graph learning and analysis tasks.

Despite the success of Lap-PEs for undirected graphs, many real-world applications involve **directed graphs**, such as circuit design, program analysis, neural architecture search, citation networks, and financial networks (Wang et al., 2020; Phothilimthana et al., 2024; Zhang et al., 2019a; Allamanis, 2022; Zhou et al., 2019; Ren et al., 2021; Liu et al., 2019; Boginski et al., 2005; Wen et al., 2020). In these graphs, the structural motifs such as bidirectional walks and loops, which consist of edges with different directions often carry critical semantic meanings. For example, in data-flow analysis, programs can be represented as data-flow graphs (Brauckmann et al., 2020; Cummins et al., 2020). Understanding reachability, liveness, and common subexpressions requires analyzing whether walks between nodes have edges pointing in the same or reverse directions (Cummins et al., 2020; 2021). Logical reasoning often depends on identifying directed relationships, such as common successors $(X \leftarrow Y \rightarrow Z)$ or predecessors $(X \rightarrow Y \leftarrow Z)$ (Qiu et al., 2023; Tian et al., 2022). Feed-forward loops that involve the two directed walks $X \rightarrow Z$ and $X \rightarrow Y \rightarrow Z$, are fundamental substructures in many biological systems (Mangan & Alon, 2003). Figure 1 shows some examples.

However, how to define PEs for directed graphs to capture the above bidirectional relations is still an open question. Most previous works focus on dealing with the asymmetry of Laplacian matrix, and propose, e.g., symmetrized Laplacian PE (Dwivedi et al., 2023), singular value decomposition PE (SVD-PE) (Hussain et al., 2022), Magnetic Laplacian PE (Mag-PE) (Geisler et al., 2023). Nevertheless, these PEs can be shown to be not expressive enough to capture the desired bidirectional relations. In this work, we address this problem by studying more expressive PEs for directed graphs. Our main contributions include:

- We formally propose a notion named *(bidirectional) walk profile*, which generalizes undirected walk counting to bidirectional walk counting and can be used to represent many important bidirectional relations in directed graphs such as directed shortest (longest) path distance, common successors/predecessors, feed-forward loops and many more. Walk profile $\Phi(m, k)$ represents the number of length-$m$ bidirectional walks with exact $k$ forward and $m - k$ backward edges.

- We show that symmetrized Laplacian PE, SVD-PE, and Mag-PE fail to express walk profile To address this problem, we propose Multi-q Magnetic Laplacian PE (Multi-q Mag-PE) that jointly takes eigenvalues and eigenvectors of multiple Magnetic Laplacian with different potential constants $q$. Interestingly, this **simple adaption** turns out to be extremely **effective**: it can provably reconstruct walk profiles of the directed graph. The key insight is that $q$ serves as the frequency of phase shift that encodes the counts of the walks with certain numbers of forward and backward edges, and using a certain number of different $q$'s allows full reconstruction of the counts of these bidirectional walks by **Fourier transform**. Notably, we show that the number of potential q have to be half of the desired walk length for perfect recovery.

- Besides, naive use of complex eigenvectors of Magnetic Laplacian could lead to severe stability issues (Wang et al., 2022). we introduce the **first** basis-invariant and stable neural architecture to handle complex eigenvectors. Specifically, we generalize the previous stable PE framework (Huang et al., 2023) **from real to complex domain**. The invariant property ensures that two equivalent complex eigenvectors (e.g., differ by a complex basis transformation) have the same representations. Moreover, the stability allows bridging Lap-PE (potential $q = 0$) and Mag-PE (potential $q \neq 0$) smoothly. Our experiments show the key role of stable PE architecture to get a good generalization performance.

- Empirical results on synthetic datasets (distance prediction, sorting network satisfiability prediction) demonstrate the stronger power of multi-q Mag-PEs to encode bidirectional relations. Real-world tasks (analog circuits prediction, high-level synthetic) show the constant performance gain of using multi-q Mag-PEs compared to existing PE methods.

## 2 RELATED WORKS

**Neural networks for directed graphs.** Neural networks for directed graphs can be mainly categorized into three types: spatial GNNs, spectral GNNs, and transformers. Spatial GNNs are those who directly use graph topology as the inductive bias in model design, including bidirectional message passing neural networks (Jaume et al., 2019; Wen et al., 2020; Kollias et al., 2022) for general

directed graphs and asynchronous message passing exclusively for directed acyclic graphs (Zhang et al., 2019b; Dong et al., 2022b; Thost & Chen, 2020). Spectral GNNs aim to generalize the concepts of Fourier basis, Fourier transform and the corresponding spectral convolution from undirected graphs to directed graphs. Potential spectral convolution operators include Magnetic Laplacian (Furutani et al., 2020; He et al., 2022; Fiorini et al., 2023; Zhang et al., 2021b), Jordan decomposition (Singh et al., 2016), Perron eigenvectors (Ma et al., 2019), motif Laplacian (Monti et al., 2018), Proximity matrix (Tong et al., 2020) and direct generalization by holomorphic functional calculus (Koke & Cremers, 2023). Finally, transformer-based models adopt the attention mechanism and their core ideas are to devise direction-aware positional encodings that better indicate graph structural information (Geisler et al., 2023; Hussain et al., 2022), which are reviewed next.

**Positional encodings for undirected/directed graphs.** Many works focus on designing positional encodings (PE) for undirected graphs, e.g., Laplacian-based PE (Dwivedi et al., 2023; Kreuzer et al., 2021; Beaini et al., 2021; Lim et al., 2022; Wang et al., 2022; Huang et al., 2023), random walk PE (Li et al., 2020; Dwivedi et al., 2021). See Rampášek et al. (2022) for a survey and Black et al. (2024) for a study that relates different types of PEs. On the other hand, recently there have been efforts to generalize PE to directed graphs. Symmetrized Laplacian symmetrizes the directed graph into an undirected one and applies regular undirected Laplacian PE (Dwivedi et al., 2023). Singular vectors of the asymmetric adjacency matrix have also been used as PE (Hussain et al., 2022; Wang et al.), and so have the eigenvectors of the Magnetic Laplacian and bidirectional random walks (Geisler et al., 2023). For directed acyclic graphs, a depth-based PE has been used (Luo et al., 2024).

## 3 PRELIMINARIES

**Basic notation.** We denote the real domain by $\mathbb{R}$ and the complex domain by $\mathbb{C}$. Bold-face letters such as $\boldsymbol{A}$ are used to denote matrices. $\boldsymbol{A}^\dagger$ denotes the conjugate transpose of $\boldsymbol{A}$. We use $i = \sqrt{-1}$ to represent the imaginary unit and $\boldsymbol{I}$ for the identity matrix.

**Directed graphs.** Let $\mathcal{G} = (\mathcal{V}, \mathcal{E})$ be a directed graph, where $\mathcal{V}$ is the node set and $\mathcal{E} \subset \mathcal{V} \times \mathcal{V}$ is the edge set. We call $u$ predecessor of $v$ if $(u, v) \in \mathcal{E}$ and call $u$ successor of $v$ if $(v, u) \in \mathcal{E}$. Let $\boldsymbol{A} \in \{0, 1\}^{n \times n}$ be the adjacency matrix of a directed graph with $n$ nodes where $\boldsymbol{A}_{u,v} = 1$ if $(u, v) \in \mathcal{E}$ and 0 otherwise. Denote $\boldsymbol{D}$ as the diagonal node degree matrix where $\boldsymbol{D}_{u,u}$ is the in-degree $d_{\text{in},u}$ plus the out-degree $d_{\text{out},u}$ of node $u$.

**Graph Magnetic Laplacian.** With a parameter $q \in \mathbb{R}$ called potential, the complex adjacency matrix $\boldsymbol{A}_q \in \mathbb{C}^{n \times n}$ is defined by $[\boldsymbol{A}_q]_{u,v} = \exp\{i2\pi q(\boldsymbol{A}_{u,v} - \boldsymbol{A}_{v,u})\}$ if $(u, v) \in \mathcal{E}$ or $(v, u) \in \mathcal{E}$, and 0 otherwise. $\boldsymbol{A}_q$ encodes edge directions via the phases of complex numbers $\exp\{\pm i2\pi q\}$. *Magnetic Laplacian $\boldsymbol{L}_q$* is correspondingly defined by $\boldsymbol{L}_q = \boldsymbol{I} - \boldsymbol{D}^{-1/2}\boldsymbol{A}_q\boldsymbol{D}^{-1/2}$. For a weighted graph $\boldsymbol{A} \in \mathbb{R}^{n \times n}$, we can instead define $[\boldsymbol{A}_q]_{u,v} = (\boldsymbol{A}_{u,v} + \boldsymbol{A}_{v,u}) \odot \exp\{i2\pi q(\boldsymbol{A}_{u,v} - \boldsymbol{A}_{v,u})\}$ and $\boldsymbol{L}_q$ thereby. As a generalization of Laplacian to directed graphs, $\boldsymbol{L}_q$ is widely studied in applied mathematics, physics and network science for directed graph analysis (Shubin, 1994; Fanuel et al., 2017; 2018), and has been recently introduced as a promising way to build graph filters and deep learning models for directed graphs (Furutani et al., 2020; Geisler et al., 2023; Fiorini et al., 2023; He et al., 2022). *Symmetrized Laplacian*, i.e., Laplacian of the undirectized graph, is essentially a special case of Magnetic Laplacian with potential $q = 0$ (no phase difference). Magnetic Laplacian $\boldsymbol{L}_q$ is hermitian since $\boldsymbol{L}_q^\dagger = \boldsymbol{L}_q$. This implies that there exists eigendecomposition $\boldsymbol{L}_q = \boldsymbol{V}\boldsymbol{\Lambda}\boldsymbol{V}^\dagger$, where $\boldsymbol{V} \in \mathbb{C}^{n \times n}$ is a unitary matrix, and $\boldsymbol{\Lambda} = \text{diag}(\boldsymbol{\lambda})$ constitutes of real eigenvalues $\boldsymbol{\lambda} \in \mathbb{R}^n$. The Magnetic Laplacian PE of node $u$ is defined by the corresponding row of $\boldsymbol{V}$, denoted by $z_u = [\boldsymbol{V}_{u,:}]^\top$.

## 4 WHAT ARE GOOD POSITIONAL ENCODINGS FOR DIRECTED GRAPHS?

In this section, we are going to study the capability of directional PEs to encode bidirectional relations. We first define a generic notion *Walk Profile*, which counts the number of walks that consist of forward or backward edges and characterizes the expressive power of PEs to represent walk profiles. As aforementioned, these bidirectional walks are important as they can form many crucial motifs to enable the studies and analysis of directed graphs, such as feed-forward loops in biological systems and common successors/predecessors for logic reasoning.

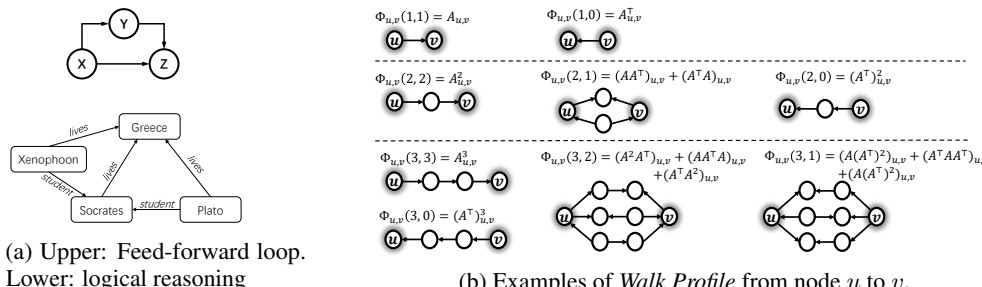

(a) Upper: Feed-forward loop.
Lower: logical reasoning

(b) Examples of *Walk Profile* from node $u$ to $v$.

Figure 1: Examples of real-world directed motifs/patterns (left) and walk profile (right).

## 4.1 Walk Profile: a General Notion that Captures Bidirectional Relations

On undirected graphs, one way to describe the spatial relation between node $u$ and $v$ is through walk counting sequences at different lengths $\mathcal{W}_{u,v} = (\boldsymbol{A}_{u,v}, \boldsymbol{A}_{u,v}^2, \boldsymbol{A}_{u,v}^3, ..., \boldsymbol{A}_{u,v}^L)$. In particular, suppose $\boldsymbol{A} \in \{0,1\}^{n \times n}$, then it is known that $\boldsymbol{A}_{u,v}^\ell$ counts the number of $\ell$-length walks from $u$ to $v$ and the shortest path distance can be found by $spd_{u,v} = \min\{\ell : \boldsymbol{A}_{u,v}^\ell > 0\}$. For directed graphs, we can adopt the same description with $\boldsymbol{A}$ replaced by the asymmetric adjacency matrix. However, this single-directional walk could miss important distance information on directed graphs. For example, consider a directed graph with $E = \{(1,3),(2,3)\}$, where node 1 and 2 share one common predecessor. However, as we cannot reach node 2 from node 1, the walk counting sequence from node 2 to node 1 is always zero. In this case this single-directional walk fail to capture the relation.

A key observation is that descriptions of directed motifs/patterns such as common successors/predecessors require knowing both the powers of $\boldsymbol{A}$ (walks via forward edges) and that of $\boldsymbol{A}^\top$ (walks via backward edges), and their combinations $\boldsymbol{A}\boldsymbol{A}^\top$ and $\boldsymbol{A}^\top\boldsymbol{A}$. This observation motivates us to define walks in a **bidirectional** manner as follows.

**Definition 4.1** (Bidirectional Walk). *Let $\mathcal{G} = (\mathcal{V}, \mathcal{E})$ be a directed graph. A bidirectional walk $w = (v_0, v_1, v_2, ...)$ is a sequence of nodes where every consecutive two nodes form either a forward edge $(v_i, v_{i+1}) \in \mathcal{E}$ or a backward edge $(v_{i+1}, v_i) \in \mathcal{E}$. The length of a bidirectional walk is the total number of forward edges and backward edges it contains.*

Bidirectional walks contain significantly more information than unidirectional walks. Note that two bidirectional walks of the same length can differ in the number of forward and backward edges. To capture this finer granularity, we introduce the concept of *Walk Profile*, a generalization of walk counting on directed graphs that retains more detailed structural information.

**Definition 4.2** (Walk Profile). *Let $\mathcal{G}$ be a directed graph and $\boldsymbol{A}$ be the adjacency matrix. Given two nodes $u, v$, Walk profile $\Phi_{u,v}(\ell, k)$ is the number of length-$\ell$ bidirectional walks from $u$ to $v$ that contains exact $k$ forward edges and $\ell - k$ backward edges.*

For example, $\Phi_{u,v}(1,1) = \boldsymbol{A}_{u,v}$ represents the connectivity from $u$ to $v$; $\Phi_{u,v}(2,1) = (\boldsymbol{A}\boldsymbol{A}^\top)_{u,v} + (\boldsymbol{A}^\top\boldsymbol{A})_{u,v}$, which counts the number 2-length walk with one forward edge and one backward edge, which corresponds to the number of common successors or predecessors. One can also compute the shortest/longest path distance using the walk profile. See Figure 1(b) for illustrations. Note that the definition of walk profile can be generalized to weighted graphs by replacing adjacency matrix with weight matrix $\boldsymbol{A} \in \mathbb{R}^{n \times n}$.

**Remark 4.1.** *Let $\mathcal{G}$ be a directed graph with Adjacency matrix $\boldsymbol{A} \in \{0,1\}^{n \times n}$, and consider two nodes $u, v$. The shortest path distance can be computed via $spd_{u,v} = \min\{\ell \in \mathbb{Z} : \Phi_{u,v}(\ell, \ell) > 0\}$. Furthermore, if $\mathcal{G}$ is acyclic, then longest path distance is $lpd_{u,v} = \max\{\ell \in \mathbb{Z} : \Phi_{u,v}(\ell, \ell) > 0\}$.*

## 4.2 The Limitations of Existing PEs to Express Walk Profile

Intuitively, a good PE can characterize certain notations of distances between nodes $u$ and $v$ based on PEs $z_u, z_v$. Now let us study the power of the existing PEs for directed graphs via the notion of walk profile. Formally, we say a PE method is expressive if it can **determine $\Phi_{u,v}$ based on $z_u$ and $z_v$**.

**Mag-PE**. Let $\boldsymbol{L}_q = \boldsymbol{I} - \boldsymbol{D}^{-1/2}\boldsymbol{A}_q\boldsymbol{D}^{-1/2}$ be Magnetic Laplacian with potential $q$. Since Magnetic graph $\boldsymbol{A}_q$ faithfully represents the directed structure of the graph, one may conjecture that eigenvalues $\lambda$ and eigenvectors $z_u, z_v$ should be able to compute the walk profile. However, it turns out that it is impossible to recover $\Phi_{u,v}(m,k)$ from them, as shown in the following Theorem 4.1.

**Theorem 4.1.** *Fix a $q \in \mathbb{R}$. There exist graphs $\mathcal{G}, \mathcal{G}'$ with adjacency matrices $\boldsymbol{A}, \boldsymbol{A}' \in \mathbb{R}^{n \times n}$, and nodes $u, v \in V_{\mathcal{G}}$ and $u', v' \in V_{\mathcal{G}'}$, such that Mag-PE $(\lambda, z_u, z_v) = (\lambda', z'_{u'}, z'_{v'})$, but $\Phi_{u,v}(m,k) \neq \Phi'_{u',v'}(m,k)$ for some $m, k$.*

**Remark 4.2.** *From the formal proof of Theorem 4.1 (see Appendix A) we can also show that Mag-PE of node $u, v$ is unable to compute shortest path distance $\mathrm{spd}_{u,v}$. More insights into why single $q$ may fail are to be discussed after the proof sketch of Theorem 4.2. Besides, as symmetrized Laplacian can be seen as a special case of Magnetic Laplacian ($q = 0$), the same negative results also apply for symmetrized Laplacian PE.*

**SVD-PE.** Singular vectors of the asymmetric adjacency matrix may also be used as directed PE (Hussain et al., 2022). That is, $\boldsymbol{A} = \boldsymbol{U}\mathrm{diag}(\boldsymbol{\sigma})\boldsymbol{W}^\top$ and define SVD-PE $z_u := (\boldsymbol{U}_{u,:}, \boldsymbol{W}_{u,:})^\top$. Here, we provide an intuition on why SVD-PE is hard to construct the walk profile. Recall that in eigen-decomposition, a power series of $\boldsymbol{A}$ can be computed via power series of eigenvalues, i.e., $f(\boldsymbol{A}) = \sum_p a_p \boldsymbol{A}^p = \boldsymbol{V}\mathrm{diag}[f(\boldsymbol{\lambda})]\boldsymbol{V}^\top$, which explains that the distance in the form of $f(\boldsymbol{A})_{u,v}$ can be expressed by $\langle \boldsymbol{V}_{u,:}, f(\boldsymbol{\lambda}) \odot \boldsymbol{V}_{v,:} \rangle$. In contrast, this property does not hold for SVD. For instance, $\boldsymbol{A}^2 = \boldsymbol{U}\mathrm{diag}(\sigma)\boldsymbol{W}^\top\boldsymbol{U}\mathrm{diag}(\sigma)\boldsymbol{W}^\top \neq \boldsymbol{U}\mathrm{diag}(\sigma^2)\boldsymbol{W}^\top$. As a result, computation of $[\boldsymbol{A}^2]_{u,v}$ (the walk profile more broadly) requires not only SVD-PEs $z_u, z_v$ but also $z_w$ for some $w \in \mathcal{V}/\{u,v\}$.

## 4.3 MAGNETIC LAPLACIAN WITH MULTIPLE POTENTIALS $q$

The limited expressivity of existing PE methods motivates us to design a more powerful direction-aware PE. The limitation of Mag-PE comes from the fact that the accumulated phase shifts over all bidirectional walks by Magnetic Laplacian cannot uniquely determine the number of walks. For example, suppose $q = 1/8$, both of the following cases - (1) node $v$ is not reachable from $u$ - v.s. - (2) $u \to u_1 \to v$ and $u \leftarrow u_2 \leftarrow v$ - yield the same $[\boldsymbol{L}_q^\ell]_{u,v} = 0$ for any integer $\ell$. However, the numbers of length-$\ell$ walks between $u$ and $v$ are different. The potential $q$ acts like a frequency that records the accumulated phase shift for a walk, and one single frequency $q$ cannot faithfully decode the distance.

Based on the frequency intepretation, we propose *Multi-q Magnetic Laplacian PE (Multi-q Mag-PE)*, which leverages multiple Magnetic Laplacians with different $q$'s simultaneously. A $q$ vector denoted by $\vec{q} = (q_1, ..., q_Q)$ is going to construct $Q$ many Magnetic Laplacian $\boldsymbol{L}_{q_1}, ..., \boldsymbol{L}_{q_Q}$, and Multi-q Mag-PE is defined by concatenation of multiple Mag-PE:

$$z_u^{\vec{q}} = ([\boldsymbol{V}_{q_1}]_{u,:}, ..., [\boldsymbol{V}_{q_Q}]_{u,:})^\top, \tag{1}$$

where $\boldsymbol{V}_{q_i}$ is the eigenvectors of $\boldsymbol{L}_{q_i}$. Intuitively, different frequencies (potential $q$) give a spectrum of phase shifts that allows to decode spatial distances in a lossless manner.

Indeed, despite the simplicity of the above extension, the following Theorem 4.2 demonstrates that this simple extension can be rather effective: with a proper number of $q$'s, one is able to exactly compute the walk profile of the desired length. Section 5.1 also provides extensive empirical justifications.

**Theorem 4.2.** *Let $L$ be a positive integer and let $Q = \lceil \frac{L}{2} \rceil + 1$, where $\lceil \cdot \rceil$ means ceiling. If we let $\vec{q} = (q_1, q_2, ..., q_Q)$ with $Q$ distinct $q$'s and $q_1, ..., q_{L+1} \in [0, \frac{1}{4})$, then for all $\ell \leq L$ and $k \leq \ell$, walk profile $\Phi_{u,v}(\ell, k)$ can be exactly computed from $(\boldsymbol{\lambda}^{\vec{q}}, z_u^{\vec{q}}, z_v^{\vec{q}})$, where $\boldsymbol{\lambda}^{\vec{q}}, z^{\vec{q}}$ are concatenation of eigenvalues/eigenvectors of different $q$ from $\vec{q}$.*

*Proof Sketch.* Fix two nodes $u, v$. As from $(\boldsymbol{\lambda}^{\vec{q}}, z_u^{\vec{q}}, z_v^{\vec{q}})$, we can construct $[\boldsymbol{A}_q^\ell]_{u,v}$ for any $q$. So the question becomes how we can determine the walk profile $\Phi_{u,v}(\cdot, \cdot)$ from $[\boldsymbol{A}_q^\ell]_{u,v}$. By the definition, one can find the following key formula that relates walk profile with $\boldsymbol{A}_q^\ell$: For any $q$,

$$[\boldsymbol{A}_q^\ell]_{u,v} = e^{-i2\pi q\ell} \sum_{k=0}^\ell \Phi_{u,v}(\ell, k) e^{i4\pi qk}. \tag{2}$$

Fix the integer $L$ and consider a length-$Q$ list of $q$ denoted by $\vec{q} = (q_1, ..., q_Q)$. Then, by Eq. 2, given $[\boldsymbol{A}_{q_1}^\ell]_{u,v}, ..., [\boldsymbol{A}_{q_Q}^\ell]_{u,v}$ for all $l \leq L$, solving $\Phi_{u,v}(\ell, k)$ is equivalent to solving a linear

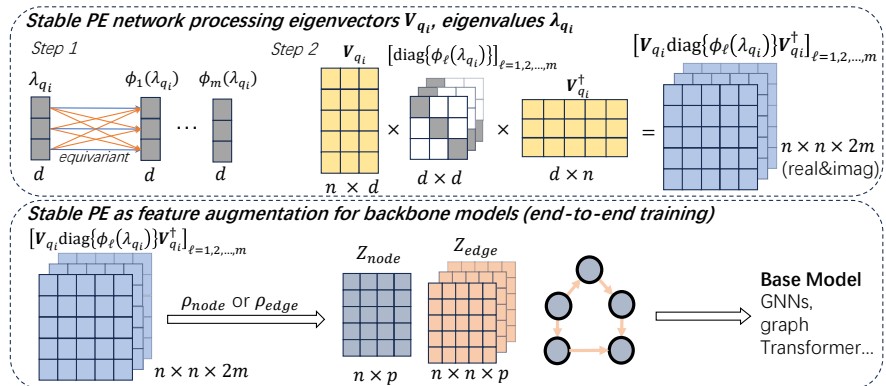

Figure 2: Multi-q Magnetic Laplacian under stable PE framework. Eigenvectors and eigenvalues of each Magnetic Laplacian with different $q$ will be processed independently and identically and concatenated in the end.

system $\boldsymbol{F\Phi} = \boldsymbol{Y}$, where $\boldsymbol{F} \in \mathbb{C}^{Q \times (L+1)}$, $\boldsymbol{\Phi} \in \mathbb{R}^{(L+1) \times L}$, $\boldsymbol{Y} \in \mathbb{C}^{Q \times L}$ and $\boldsymbol{F}_{j,m} = \exp\{i4\pi q_j m\}$, $\boldsymbol{\Phi}_{j,m} = \Phi_{u,v}(m,j)$ (equals zero if $j > m$), $\boldsymbol{Y}_{j,m} = [\boldsymbol{A}_{q_j}^m]_{u,v} \cdot e^{i2\pi q_j m}$. In order to make the linear system $\boldsymbol{F}$ well-posed for uniquely solving the walk profile $\boldsymbol{\Phi}$ from $\boldsymbol{Y}$, it generally requires $\boldsymbol{F}$ to have $Q = L + 1$ distinct rows, i.e., $e^{i4\pi q_i} \neq e^{i4\pi q_j}$ for any $i \neq j$, such that Fourier matrix $\boldsymbol{F}$ becomes full-rank. Fortunately, as the unknowns $\boldsymbol{\Phi}$ lies in real space, the number of rows $Q$ can be reduced to $\lceil L/2 \rceil + 1$ thanks to the symmetry of Fourier coefficients $\boldsymbol{Y}$ of real $\boldsymbol{\Phi}$. As a result, we can choose $Q = \lceil L/2 \rceil + 1$ many distinct qs with $q_i \in [0, \frac{1}{4})$, by which we can construct a full-rank system to solve $\boldsymbol{\Phi}$. $\qquad\square$

The proof provides several insights: (a) determining the walk profile from the powers of $\boldsymbol{A}_q$ of a **single** $q$ is **ill-posed**; (b) in contrast, using $\lceil L/2 \rceil + 1$ many $q$'s simultaneously makes the problem well-posed and allows it to uniquely determine the walk profile of length $\leq L$. The number of $q$ can be chosen based on the walk length of interest for a specific task; (c) the capability of expressing the walk profile is robust to the values of multiple $q$'s. In practice, we may choose evenly-spaced $\vec{q} = (\frac{0}{2(L+1)}, ..., \frac{\lceil L/2 \rceil + 1}{2(L+1)})$ where $\boldsymbol{F}$ indicates **discrete Fourier transform**, or randomly sampled $q_i \in [0, \frac{1}{4})$ where $\boldsymbol{F}$ is non-singular with high probability.

**Limitations.** Despite the provable effectiveness of Multi-q Mag-PE, it suffers from some limitations. One is its extra computational overhead: PEs induced by multiple $q$'s require multiple runs of eigenvalue decomposition and introduce the higher PE input dimension. This introduces a trade-off between the expressivity of PEs and computational complexity, which can be tuned in practice by selecting different numbers of $q$'s. We provide a runtime comparison with various numbers of $q$'s, see Sec. 5.4. Typically, the actual runtime of $Q = 5$ is about twice of the runtime when $Q = 1$. We also discuss the pros and cons of using Multi-q Mag-PE v.s. directly using walk profiles as injected features in Appendix G.

### 4.4 THE FIRST BASIS-INVARIANT/STABLE NEURAL ARCHITECTURE FOR COMPLEX EIGENVECTORS

Recent studies have shown that Lap-PE has the issues of basis ambiguity and instability (Wang et al., 2022; Lim et al., 2022; Huang et al., 2023; Black et al., 2024). That is, because eigendecomposition is not unique (i.e., $\boldsymbol{L} = \boldsymbol{V\Lambda V}^\top = (\boldsymbol{VQ})\boldsymbol{\Lambda}(\boldsymbol{VQ})^\top$ for some orthogonal matrix $\boldsymbol{Q}$), Lap-PE can become completely different for the same Laplacian and tend to be unstable to Laplacian perturbation. In fact, this problem technically becomes even harder for Mag-PEs because Mag-PEs live in the complex domain and the basis ambiguity extends to unitary basis transform: $\boldsymbol{L}_q = \boldsymbol{V\Lambda V}^\dagger = (\boldsymbol{VQ})\boldsymbol{\Lambda}(\boldsymbol{VQ})^\dagger$ for some unitary matrix $\boldsymbol{Q} \in \mathbb{C}^{n \times n}$. For Multi-q Mag-PE, even without duplicated eigenvalues, eigenvectors associated with each $q$ exhibit their own symmetry (if $v$ is an eigenvector, $ve^{i\theta}$ for any $\theta \in (0, 2\pi)$ is also an eigenvector), exacerbating the ambiguity and stability issues even further.

To address this problem, we aim to generalize the previous stable PE frameworks, PEG (Wang et al., 2022) and SPE (Huang et al., 2023), to handle complex eigenvectors. Our framework will process PEs into stable representations and use them as augmented node/edge features in the backbone model. We first consider Mag-PE of a single $q$ and then extend it. Specifically, let $\phi_\ell^{\text{node}}, \phi_\ell^{\text{edge}} : \mathbb{R}^d \to \mathbb{R}^d$ be permutation-equivariant function w.r.t. $d$-dim axis (i.e., equivariant to permutation of eigenvalues), we can construct node-level stable PE $z_{\text{node}} \in \mathbb{R}^{n \times d}$ and/or edge-level stable PE $z_{\text{edge}} \in \mathbb{R}^{n \times n \times d}$:

$$
\begin{aligned}
z_{\text{node}} = \rho_{\text{node}}(&\text{Re}\{\boldsymbol{V}\text{diag}(\phi_1^{\text{node}}(\boldsymbol{\lambda}))\boldsymbol{V}^\dagger\}, ..., \text{Re}\{\boldsymbol{V}\text{diag}(\phi_m^{\text{node}}(\boldsymbol{\lambda}))\boldsymbol{V}^\dagger\}, \\
&\text{Im}\{\boldsymbol{V}\text{diag}(\phi_1^{\text{node}}(\boldsymbol{\lambda}))\boldsymbol{V}^\dagger\}, ..., \text{Im}\{\boldsymbol{V}\text{diag}(\phi_m^{\text{node}}(\boldsymbol{\lambda}))\boldsymbol{V}^\dagger\}),
\end{aligned}
\tag{3}
$$

$$
\begin{aligned}
z_{\text{edge}} = \rho_{\text{edge}}(&\text{Re}\{\boldsymbol{V}\text{diag}(\phi_1^{\text{edge}}(\boldsymbol{\lambda}))\boldsymbol{V}^\dagger\}, ..., \text{Re}\{\boldsymbol{V}\text{diag}(\phi_m^{\text{edge}}(\boldsymbol{\lambda}))\boldsymbol{V}^\dagger\}, \\
&\text{Im}\{\boldsymbol{V}\text{diag}(\phi_1^{\text{edge}}(\boldsymbol{\lambda}))\boldsymbol{V}^\dagger\}, ..., \text{Im}\{\boldsymbol{V}\text{diag}(\phi_m^{\text{edge}}(\boldsymbol{\lambda}))\boldsymbol{V}^\dagger\}),
\end{aligned}
\tag{4}
$$

where $\boldsymbol{V}, \boldsymbol{\lambda}$ are the eigenvectors and eigenvalues of Magnetic Laplacian with a certain $q$, $\text{Re}\{\cdot\}, \text{Im}\{\cdot\}$ means taking the real and imaginary parts, respectively, and $\rho_{\text{node}} : \mathbb{R}^{n \times n \times 2m} \to \mathbb{R}^{n \times p}$ and $\rho_{\text{edge}} : \mathbb{R}^{n \times n \times 2m} \to \mathbb{R}^{n \times n \times p}$ are permutation equivariant function w.r.t. $n$-dim axis (i.e., equivariant to permutation of node indices). Afterwards, $z_{\text{node}}$ will be concatenated with node features, and $z_{\text{edge}}$ will be concatenated with edge features.

Note that this is the first work to propose the usage of complex PEs in a stable way. In practice, if the backbone model is a GNN, only a portion of (sparse) entries $[z_{\text{edge}}]_{u,v}, (u,v) \in \mathcal{E}$ need to be computed. For the multi-$q$ case, we apply the same $\phi_\ell$ and $\rho$ to the eigenvectors and eigenvalues from different $q$'s and concatenate the outputs. A similar proof technique can show that the above stable PE framework can achieve generalization benefits as stated in Proposition 3.1 in Huang et al. (2023). Moreover, it can be shown that $z_{\text{node}}, z_{\text{edge}}$ are continuous in the choice of $q$ due to the stable structure. Such continuity naturally unifies Lap-PE and Mag-PE, because symmetrized Laplacian $\boldsymbol{L}_s$ is a special case of Magnetic Laplacian $\boldsymbol{L}_q$ when $q \to 0$.

## 5 EXPERIMENTS

In this section, we evaluate the effectiveness of multi-q Mag-PEs by studying the following questions:
- **Q1**: How good are the previous PEs and our proposed PEs at expressing directed distances/relations, e.g., directed shortest/longest path distances and the walk profile?
- **Q2**: How do these PE methods perform on practical tasks and real-world datasets?
- **Q3**: What is the impact on using PEs with or without basis-invariant/stable architectures?

In our experiments, we mainly consider three ways of processing PEs: (1) Naïve: directly concatenates raw PEs with node features; (2) SignNet (Lim et al., 2022): makes PEs sign invariant. We adopt the same pipeline as in (Geisler et al., 2023), Figure G.1; (3) SPE (Eqs. 3,4): we follow (Huang et al., 2023), and use element-wise MLPs as $\phi_1, ..., \phi_m$, GIN (Xu et al., 2018) as $\rho_{\text{node}}$ and MLPs as $\rho_{\text{edge}}$. Key hyperparameters are included in the main text while full details of the experiment setup and model configurations can be found in Appendix B.

### 5.1 DISTANCE PREDICTION ON DIRECTED GRAPHS

**Datasets.** To answer question **Q1**, we follow Geisler et al. (2023) and generate Erdős–Rényi random graphs. Specifically, we sample regular directed graphs with average node degree drawn from $\{1, 1.5, 2\}$, or directed acyclic graphs with average node degree from $\{1, 1.5, 2, 2.5, 3\}$. In both cases, there are 400,000 samples for training and validation (graph size from 16 to 63, training:validation=95:5), and 5,000 samples for test (graph size from 64 to 71). Finally, We take the largest connected component of each generated graph and form our final dataset. The task is to predict the pair-wise distances for node pairs measured by: (1) shortest path distance; (2) longest path distance; (3) walk profile. Only node pairs that are reachable or have non-zero walk profile elements are included for training and test. For the walk profile, we choose to predict the normalized walk profile of length 4, which is a 5-dim vector. Normalized walk profiles corresponds to the probability of bidirectional random walks, with adjacency matrix $\boldsymbol{A}$ in walk profiles replaced by random walk $\boldsymbol{D}_{\text{total}}^{-1}\boldsymbol{A}$, as the latter is consistent with the normalized Laplacian $\boldsymbol{I} - \boldsymbol{D}_{\text{total}}^{-1/2}\boldsymbol{A}\boldsymbol{D}_{\text{total}}^{-1/2}$ we use in practice. See Appendix H for discussions.

Table 1: Test RMSE results over 3 random seeds for node-pair distance prediction.

| PE method | PE processing | Directed Acyclic Graph | | | Regular Directed Graph | | |
|---|---|---|---|---|---|---|---|
| | | $spd$ | $lpd$ | $wp(4,\cdot)$ | $spd$ | $lpd$ | $wp(4,\cdot)$ |
| Lap | Naive | $0.488_{\pm0.005}$ | $0.727_{\pm0.005}$ | $0.370_{\pm0.004}$ | $2.068_{\pm0.004}$ | $1.898_{\pm0.001}$ | $0.480_{\pm0.000}$ |
| | SignNet | $0.537_{\pm0.013}$ | $0.771_{\pm0.013}$ | $0.437_{\pm0.000}$ | $2.064_{\pm0.004}$ | $1.900_{\pm0.002}$ | $0.518_{\pm0.027}$ |
| | SPE | $0.355_{\pm0.001}$ | $0.655_{\pm0.002}$ | $0.326_{\pm0.001}$ | $2.066_{\pm0.005}$ | $1.920_{\pm0.000}$ | $0.452_{\pm0.001}$ |
| SVD | Naive | $0.649_{\pm0.002}$ | $0.853_{\pm0.002}$ | $0.721_{\pm0.000}$ | $2.196_{\pm0.002}$ | $1.982_{\pm0.004}$ | $0.519_{\pm0.000}$ |
| | SignNet | $0.673_{\pm0.003}$ | $0.872_{\pm0.002}$ | $0.443_{\pm0.001}$ | $2.229_{\pm0.003}$ | $1.996_{\pm0.005}$ | $0.541_{\pm0.001}$ |
| | SPE | $0.727_{\pm0.001}$ | $0.912_{\pm0.001}$ | $0.721_{\pm0.000}$ | $2.261_{\pm0.002}$ | $2.122_{\pm0.007}$ | $0.755_{\pm0.000}$ |
| MagLap-1q (q=0.1) | Naive | $0.366_{\pm0.003}$ | $0.593_{\pm0.003}$ | $0.241_{\pm0.010}$ | $1.826_{\pm0.005}$ | $1.760_{\pm0.007}$ | $0.311_{\pm0.013}$ |
| | SignNet | $0.554_{\pm0.001}$ | $0.699_{\pm0.002}$ | $0.268_{\pm0.042}$ | $2.048_{\pm0.005}$ | $1.881_{\pm0.003}$ | $0.413_{\pm0.001}$ |
| | SPE | $0.124_{\pm0.002}$ | $0.433_{\pm0.002}$ | $0.043_{\pm0.000}$ | $1.620_{\pm0.005}$ | $1.547_{\pm0.004}$ | $0.133_{\pm0.001}$ |
| MagLap-1q (best q) | SPE | $0.124_{\pm0.002}$ | $0.432_{\pm0.004}$ | $0.040_{\pm0.001}$ | $1.533_{\pm0.007}$ | $1.493_{\pm0.003}$ | $0.132_{\pm0.000}$ |
| MagLap-Multi-q | Naive | $0.353_{\pm0.003}$ | $0.535_{\pm0.006}$ | $0.188_{\pm0.010}$ | $1.708_{\pm0.012}$ | $1.661_{\pm0.002}$ | $0.257_{\pm0.007}$ |
| | SignNet | $0.473_{\pm0.000}$ | $0.579_{\pm0.001}$ | $0.280_{\pm0.004}$ | $1.906_{\pm0.001}$ | $1.784_{\pm0.008}$ | $0.377_{\pm0.007}$ |
| | SPE | $\mathbf{0.016_{\pm0.000}}$ | $\mathbf{0.185_{\pm0.036}}$ | $\mathbf{0.002_{\pm0.000}}$ | $\mathbf{0.546_{\pm0.068}}$ | $\mathbf{1.100_{\pm0.007}}$ | $\mathbf{0.074_{\pm0.001}}$ |

| PE method | PE processing | Test F1 |
|---|---|---|
| Lap | Naive | $51.39_{\pm2.36}$ |
| | SignNet | $49.50_{\pm4.01}$ |
| | SPE | $56.35_{\pm3.70}$ |
| Maglap-1q (best q) | Naive | $68.68_{\pm9.66}$ |
| | SignNet | $76.28_{\pm6.82}$ |
| | SPE | $86.86_{\pm3.85}$ |
| Maglap-5q | Naive | $75.12_{\pm12.78}$ |
| | SignNet | $72.97_{\pm9.38}$ |
| | SPE | $\mathbf{91.27_{\pm0.71}}$ |

Table 2: Test F1 scores over 5 random seeds for sorting network satisfiability.

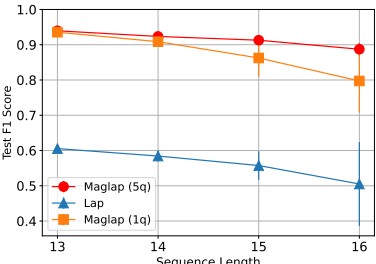

Figure 3: Test F1 scores w.r.t. different sorting network lengths.

**Models.** To show the power of pure PEs, we take PEs $z_u, z_v$ of node $u, v$ to predict the distance $d_{u,v}$ between $u, v$. We consider three ways of processing PEs: (1) naïve concatenation: $\text{MLP}([z_u, z_v])$; (2) SignNet-based: $\text{MLP}(\text{SignNet}(z_u), \text{SignNet}(z_v))$; (3) edge-SPE-based: $\text{MLP}([z_{\text{edge}}]_{u,v})$ where $z_{\text{edge}}$ is defined in Eq. 4. For Multi-q Magnetic Laplacian PE, we choose $\vec{q} = (0, \frac{1}{2L}, ..., \frac{L-1}{2L})$ where $L = 5$ for predicting walk profile, $L = 10$ for predicting shortest/longest path distances on directed acyclic graphs and $L = 15$ for predicting shortest/longest path distances on regular directed graphs. For single-q Mag-PE, we tune $q$ from 0 to 0.5, and we report both the result of $q = 0.1$ and the tuned best $q$. It turns out the performance of single-q Mag-PE in this task is not sensitive to the value of $q$ as long as not being too large.

**Results.** Table 1 indicates several conclusions. (1) Multi-q Mag-PE constantly outperforms other PE methods regardless of how we process the PE. Particularly, it is significantly better than other methods when equipped with SPE; (2) *impact of symmetry:* when restricted to the use of naïve concatenation, the overall performance of all PE methods is not desired. This is because naive concatenation cannot handle the basis ambiguity of eigenvectors. It is also worth noticing that SignNet's performance may be even worse than naïve concatenation. This is because the PE processed by SignNet suffers from node ambiguity, which loses pair-wise distance information (Zhang et al., 2021a; Lim et al., 2023). Besides, SignNet is not stable. In contrast, when we properly handle the complex eigenvectors by SPE, the true benefit of Mag-PE starts to present. On average, the test RMSE of Multi-q Magnetic Laplacian gets reduced by $66\%$ compared to the one uses naïve concatenation.

**Ablation study.** The comparison to single-q Mag-PE naturally serves as an ablation study of Multi-q Mag-PE. One may wonder if Multi-q Mag-PE performs better because it can search and find the best $q$ from $\vec{q}$. We can show that the joint use of different $q$'s is indeed necessary, and it is constantly better than using the single best $q$. The hyper-parameter search of the best $q$ for single-q Magnetic Laplacian for different tasks can be found in Appendix C.

**Robustness of multiple $q$ values.** As implied by Theorem 4.2, the expressivity to compute walk profile is robust to the choice of multiple $q$ values. To verify this, Multi-q Mag-PE with a randomly sampled $\vec{q}$ (draw each $q$ from uniform distribution) is implemented. See details in Appendix E. Our results show that random $\vec{q}$ sometimes achieves even better prediction than evenly-spaced $\vec{q}$.

Table 3: Test results (RMSE for Gain/BW/PM, MSE for DSP/LUT) for Open Circuit Benchmark and High-level Synthesis. For each base model and task, **Bold** denotes the best result for each base model and task, and **Bold*** for the best result that exceeds the second best result by one standard deviation.

| Base Model | PE method | PE processing | Gain | BW | PM | DSP | LUT |
|---|---|---|---|---|---|---|---|
| Undirected-GIN | None | N/A | $0.416_{\pm0.017}$ | $4.908_{\pm0.083}$ | $1.119_{\pm0.011}$ | $2.827_{\pm0.206}$ | $2.153_{\pm0.169}$ |
| | Lap | SignNet | $0.430_{\pm0.009}$ | $4.712_{\pm0.134}$ | $1.127_{\pm0.007}$ | $2.665_{\pm0.184}$ | $2.025_{\pm0.057}$ |
| | | SPE | $0.416_{\pm0.021}$ | $4.321_{\pm0.084}$ | $1.127_{\pm0.020}$ | $2.662_{\pm0.187}$ | $\mathbf{1.925_{\pm0.059}}$ |
| | Maglap-1q (q=0.01) | SignNet | $0.426_{\pm0.009}$ | $4.670_{\pm0.113}$ | $1.116_{\pm0.009}$ | $2.673_{\pm0.090}$ | $2.027_{\pm0.091}$ |
| | | SPE | $0.405_{\pm0.016}$ | $4.305_{\pm0.092}$ | $1.121_{\pm0.018}$ | $2.666_{\pm0.190}$ | $2.024_{\pm0.068}$ |
| | Maglap-1q (best q) | SPE | $0.398_{\pm0.025}$ | $4.281_{\pm0.085}$ | $\mathbf{1.113_{\pm0.022}}$ | $2.614_{\pm0.098}$ | $2.010_{\pm0.082}$ |
| | Maglap-Multi-q | SignNet | $0.421_{\pm0.015}$ | $4.743_{\pm0.215}$ | $1.126_{\pm0.011}$ | $2.665_{\pm0.111}$ | $2.025_{\pm0.076}$ |
| | | SPE | $\mathbf{0.389_{\pm0.017}}$ | $\mathbf{4.175^*_{\pm0.115}}$ | $1.137_{\pm0.004}$ | $\mathbf{2.582_{\pm0.133}}$ | $1.976_{\pm0.089}$ |
| Bidirected-GIN | None | N/A | $0.386_{\pm0.008}$ | $4.594_{\pm0.087}$ | $1.123_{\pm0.010}$ | $2.232_{\pm0.143}$ | $1.939_{\pm0.068}$ |
| | Lap | SignNet | $0.382_{\pm0.008}$ | $4.371_{\pm0.171}$ | $1.127_{\pm0.021}$ | $2.256_{\pm0.109}$ | $1.806_{\pm0.096}$ |
| | | SPE | $0.391_{\pm0.007}$ | $4.153_{\pm0.160}$ | $1.135_{\pm0.035}$ | $2.267_{\pm0.126}$ | $1.786_{\pm0.072}$ |
| | Maglap-1q (q=0.01) | SignNet | $0.388_{\pm0.012}$ | $4.351_{\pm0.132}$ | $1.131_{\pm0.021}$ | $2.304_{\pm0.143}$ | $1.882_{\pm0.085}$ |
| | | SPE | $0.384_{\pm0.008}$ | $4.152_{\pm0.056}$ | $1.123_{\pm0.026}$ | $2.344_{\pm0.134}$ | $1.830_{\pm0.116}$ |
| | Maglap-1q (best q) | SPE | $0.383_{\pm0.002}$ | $4.113_{\pm0.052}$ | $\mathbf{1.099_{\pm0.020}}$ | $2.256_{\pm0.144}$ | $1.768_{\pm0.090}$ |
| | Maglap-Multi-q | SignNet | $0.381_{\pm0.008}$ | $4.443_{\pm0.116}$ | $1.119_{\pm0.016}$ | $2.212_{\pm0.116}$ | $1.791_{\pm0.091}$ |
| | | SPE | $\mathbf{0.371^*_{\pm0.008}}$ | $\mathbf{4.051^*_{\pm0.139}}$ | $1.116_{\pm0.012}$ | $\mathbf{2.207_{\pm0.185}}$ | $\mathbf{1.735_{\pm0.096}}$ |
| SAT (undirected-GIN) | None | N/A | $0.368_{\pm0.019}$ | $4.107_{\pm0.103}$ | $1.077_{\pm0.032}$ | $3.154_{\pm0.263}$ | $2.286_{\pm0.147}$ |
| | Lap | SignNet | $0.368_{\pm0.022}$ | $4.085_{\pm0.189}$ | $1.038_{\pm0.016}$ | $3.103_{\pm0.101}$ | $2.223_{\pm0.175}$ |
| | | SPE | $0.375_{\pm0.016}$ | $4.180_{\pm0.093}$ | $1.065_{\pm0.034}$ | $3.167_{\pm0.193}$ | $2.425_{\pm0.168}$ |
| | Maglap-1q (q=0.01) | SignNet | $0.382_{\pm0.014}$ | $4.143_{\pm0.181}$ | $1.073_{\pm0.021}$ | $3.087_{\pm0.183}$ | $2.214_{\pm0.150}$ |
| | | SPE | $0.366_{\pm0.003}$ | $4.081_{\pm0.071}$ | $1.089_{\pm0.023}$ | $3.206_{\pm0.197}$ | $2.362_{\pm0.154}$ |
| | Maglap-1q (best q) | SPE | $0.361_{\pm0.016}$ | $\mathbf{4.014_{\pm0.068}}$ | $1.057_{\pm0.036}$ | $3.101_{\pm0.176}$ | $2.362_{\pm0.154}$ |
| | Maglap-Multi-q | SignNet | $0.368_{\pm0.020}$ | $4.044_{\pm0.090}$ | $1.066_{\pm0.028}$ | $3.121_{\pm0.143}$ | $\mathbf{2.207_{\pm0.113}}$ |
| | | SPE | $\mathbf{0.350_{\pm0.004}}$ | $4.044_{\pm0.153}$ | $\mathbf{1.035_{\pm0.025}}$ | $\mathbf{3.076_{\pm0.240}}$ | $2.333_{\pm0.147}$ |
| SAT (bidirected-GIN) | None | N/A | $0.392_{\pm0.036}$ | $4.035_{\pm0.111}$ | $1.065_{\pm0.019}$ | $2.724_{\pm0.158}$ | $2.117_{\pm0.106}$ |
| | Lap | SignNet | $0.384_{\pm0.025}$ | $3.949_{\pm0.125}$ | $1.069_{\pm0.029}$ | $\mathbf{2.569_{\pm0.116}}$ | $2.048_{\pm0.088}$ |
| | | SPE | $0.368_{\pm0.022}$ | $4.024_{\pm0.106}$ | $1.046_{\pm0.021}$ | $2.713_{\pm0.135}$ | $2.173_{\pm0.107}$ |
| | Maglap-1q (q=0.01) | SignNet | $0.384_{\pm0.015}$ | $4.023_{\pm0.032}$ | $1.055_{\pm0.028}$ | $2.616_{\pm0.120}$ | $2.054_{\pm0.127}$ |
| | | SPE | $0.364_{\pm0.012}$ | $3.996_{\pm0.178}$ | $1.074_{\pm0.030}$ | $2.687_{\pm0.209}$ | $2.192_{\pm0.135}$ |
| | Maglap-1q (best q) | SPE | $0.360_{\pm0.009}$ | $3.960_{\pm0.060}$ | $1.062_{\pm0.024}$ | $2.657_{\pm0.128}$ | $2.107_{\pm0.135}$ |
| | Maglap-Multi-q | SignNet | $0.420_{\pm0.035}$ | $4.022_{\pm0.128}$ | $1.089_{\pm0.046}$ | $2.741_{\pm0.110}$ | $\mathbf{2.045_{\pm0.079}}$ |
| | | SPE | $\mathbf{0.359_{\pm0.008}}$ | $\mathbf{3.930_{\pm0.069}}$ | $\mathbf{1.045_{\pm0.012}}$ | $2.616_{\pm0.151}$ | $2.082_{\pm0.099}$ |

## 5.2 SORTING NETWORK SATISFIABILITY

Sorting network (Knuth, 1997) is a comparison-based algorithm designed to perform sorting on a fixed number of variables. Each sorting network consists of a sequence of operations $v_i, v_j = sorted(v_i, v_j)$ and we say a sorting network is satisfiable if it can correctly sort an arbitrary input of a given length. It can be parsed into a directed graph for which the direction (i.e., the order of operations) impacts the satisfiability. We test the ability of positional encodings by how well they can predict the satisfiability.

**Datasets.** We randomly generate sorting networks by following the setup from Geisler et al. (2023). Sorting networks are parsed into directed acyclic graphs whose nodes are comparison operators and directed edges connect two operators that share variables to sort. The dataset contains 800k training samples with a length (the number of variables to sort) from 7 to 11, 60k validation samples with a length 12, and 60k test samples with a length from 13 to 16. We perform graph classification to predict satisfiability.

**Models.** We adopt a vanilla transformer (Vaswani et al., 2017) as our base model. PEs are either naively concatenated into node features, or using SignNet before concatenation, or concatenated both into node features and attention weights using SPEs Eqs.3 and 4. For single-q Magnetic Laplacian PE, we tune $q$ from $\frac{1}{20 \cdot d_G}$ to $\frac{5}{20 \cdot d_G}$, where where $d_G = \max(\min(m, n), 1)$, $m$ is the number of directed edges and $n$ is the number of nodes. We find $q = \frac{5}{20 \cdot d_G}$ gives the best results, which is the same $q$ as in Geisler et al. (2023). For multi-q Mag-PE, we choose $\vec{q} = (\frac{1}{20 \cdot d_G}, ..., \frac{5}{20 \cdot d_G})$.

**Results.** Table 2 displays the average Test F1 score of classifying satisfiability. Again, Multi-q Mag-PE equipped with SPE achieves the best performance. Mag-PE with SignNet (both single-q and multi-q) performs poorly compared to their SPE counterparts. Figure 3 additionally illustrates the test F1 on samples with respect to individual sorting network lengths. Note that although single-q Mag-PE and multi-q Mag-PE perform equally well on length=13 samples, the single-q one generalizes worse on longer-length sorting networks. In contrast, Multi-q Mag-PE has nearly the same generalization performance as the number of variables to sort increases.

### 5.3 CIRCUIT PROPERTY PREDICTION

**Open Circuit Benchmark.** Open Circuit Benchmark (Dong et al., 2022a) contains 10,000 operational amplifiers circuits as directed graphs and the task is to predict the DC gain (Gain), band width (BW) and phase margin (PM) of each circuit. These targets reflect the property of current flows from input nodes to output nodes and thus require a powerful direction-aware model. The dataset consists of 2-stage amplifiers and 3-stage amplifiers and we use 2-stage amplifiers in our experiment. We randomly split them into 0.9:0.05:0.05 as training, validation and test set.

**High-level Synthesis.** The HLS dataset (Wu et al., 2022) collects 18,750 intermediate representation (IR) graphs of C/C++ code after front-end compilation (Alfred et al., 2007). It provides post-implementation performance metrics on FPGA devices as labels, which are obtained after hours of synthesis using the Vitis HLS tool (vit) and implementation with Vivado (viv). The task is to predict resource usage, namely look-up table (LUT) and digital signal processor (DSP) usage. We randomly select 16570 for training, and 1000 each for validation and testing.

**Models.** We adopt GIN (Xu et al., 2018) as the backbone and implement two variants: (1) undirected-GIN: the normal GIN works on the undirected version of the original directed graphs; (2) bidirectional-GIN: bidirectional message passing with different weights for two directions, inspired by (Jaume et al., 2019; Thost & Chen, 2020; Wen et al., 2020). The state-of-the-art graph transformer SAT (Chen et al., 2022) is also adopted, whose GNN extractor is undirected-GIN or bidirected-GIN as mentioned for self-attention computation. PEs are processed and then concatenated with node features using SignNet or with node and edge features using SPE (Eqs. 3, 4). We generally choose $\vec{q} = (1, 2, ..., 10)/100$ or $\vec{q} = (1, 2, ..., 5)/100$ for Multi-q Mag-PE (see Appendix B for specific $q$ for each tasks). For single-q Magnetic Laplacian, we report $q = 0.01$ as well as the best results of single $q$ by searching over the range of multiple $q$. See Appendix C for full results of varying single $q$.

**Results.** Table 3 shows the test RMSE (5 random seeds) on Open Circuit Benchmark (Gain, BW, PM) and the test MSE (10 random seeds) on high-level synthesis (DSP, LUT). Notably, Multi-q Mag-PE with Stable PE framework achieves most of the best results for 5 targets compared to other PEs.

### 5.4 RUNTIME EVALUATION

Increasing the number of $q$ boosts expressivity but brings extra computational costs. We evaluate the runtime of preprocessing (eigendecomposition), training and inference for various number of $q$, as shown in Figure 4. We can see that the training time of even 10 $q$ is generally about 1.5 to 3 times the training time of one single $q$. The preprocessing time is not a concern as they are negligible compared to total training time for hundreds of epochs in our experiments. See detailed setup at Appendix D.

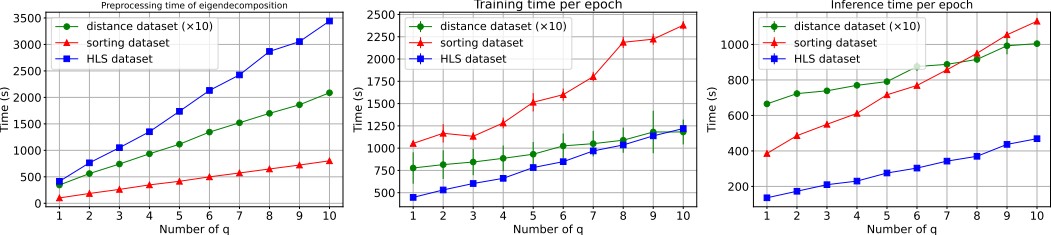

Figure 4: Runtime of preprocessing (left), training (middle) and inference (right) on three datasets. The shown runtime of distance dataset is 10 times the actual ones for better illustration.

## 6 CONCLUSION

This work studies the positional encodings (PEs) for directed graphs. We propose the notion of walk profile to assess the model's ability to encode directed relation. Limitations of existing PEs to express walk profiles are identified. We propose a simple yet effective variant of Magnetic Laplacian PE called Multi-q Mag-PE that can provably compute walk profile. the basis-invariant and stable PE framework is extened to address the basis ambiguity and stability problem of complex eigenvectors. Experiments demonstrate the consistent performance gain of Multi-q Mag-PE.

ACKNOWLEDGMENTS

This research is supported by the NSF awards PHY-2117997, IIS-2239565, CFF-2402816, IIS-2428777 and JPMorgan Faculty award.

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

# A DEFERRED PROOFS

## A.1 PROOF OF THEOREM 4.1

**Theorem 4.1.** *Fix a $q \in \mathbb{R}$. There exist graphs $\mathcal{G}, \mathcal{G}'$ with adjacency matrices $\boldsymbol{A}, \boldsymbol{A}' \in \mathbb{R}^{n \times n}$, and nodes $u, v \in V_{\mathcal{G}}$ and $u', v' \in V_{\mathcal{G}'}$, such that Mag-PE $(\lambda, z_u, z_v) = (\lambda', z'_{u'}, z'_{v'})$, but $\Phi_{u,v}(m, k) \neq \Phi'_{u',v'}(m, k)$ for some $m, k$.*

*Proof.* Fix a $q \in \mathbb{R}$ and two nodes $u, v$. Pick and fix anther node $u_0$ that is not $u, v$. Let us define a graph $\boldsymbol{A} \in \mathbb{R}^{n \times n}$ such that $\text{diag}(\boldsymbol{A}) = 0$, and for all $w, r \in \mathcal{V}$, $\boldsymbol{A}_{u_0,w} = \boldsymbol{A}_{u_0,r} = 1$ and $\boldsymbol{A}_{w,r} = 0$ otherwise. for all nodes $w, r$. Note that complex adjacency matrix $\boldsymbol{A}_q = (\boldsymbol{A} + \boldsymbol{A}^\top) \odot \exp\{i2\pi q(\boldsymbol{A} - \boldsymbol{A}^\top)\}$ uniquely determines $\boldsymbol{A}$ by the relation $\boldsymbol{A} = \frac{1}{2}|\boldsymbol{A}_q| + \frac{1}{2}\frac{\angle \boldsymbol{A}_q}{2\pi q}$, where $|\cdot|, \angle$ denote amplitude and phase of complex numbers.

Now let us construct $\boldsymbol{A}'$ by constructing $\boldsymbol{A}'_q$ first. Suppose Hermitian $\boldsymbol{A}_q$ has eigenvalue decomposition $[\boldsymbol{A}_q]_{w,r} = \sum_k \lambda_k z_{w,k} z^*_{r,k}$. Let us define another group of eigenvalues $\lambda'$ and eigenvectors (PE) $z'$ via:

$$\lambda' = \lambda, \quad z'_{w,:} = \begin{cases} z_{w,:}, & \text{if } w \neq u_0 \\ e^{i\theta} z_{u_0,:} & w = u_0 \end{cases}, \tag{5}$$

that is, $\lambda', z'$ shares the exact same spectrum as $\lambda, z$, except that $z'_{u_0,:} = \exp(i\theta) \cdot z_{u_0,:}$. It is easy to verify that $z'$ is indeed eigenvectors by their orthonormality, so we can construct Hermitian $\boldsymbol{A}'_q$ as $[\boldsymbol{A}'_q]_{w,r} = \sum_k \lambda'_k z'_{w,k} z'^*_{r,k}$. From this construction, we shall see that $(\lambda, z_u, z_v) = (\lambda', z'_u, z'_v)$.

On the other hand, let us examine the walk profile. For any two nodes $w, r$, we have

$$[\boldsymbol{A}'_q]_{w,r} = \sum_k \lambda'_k z'_{w,k} z'^*_{r,k} = \begin{cases} [\boldsymbol{A}_q]_{w,r}, & \text{if } w, r \neq u_0 \\ e^{-i\theta}[\boldsymbol{A}_q]_{w,u_0}, & \text{if } w \neq u_0, r = u_0 \\ e^{i\theta}[\boldsymbol{A}_q]_{u_0,r}, & \text{if } w = u_0, r \neq u_0 \end{cases} \tag{6}$$

This determines $\boldsymbol{A}'$, which reads

$$[\boldsymbol{A}']_{w,r} = \frac{1}{2}|[\boldsymbol{A}'_q]_{w,r}| + \frac{1}{2}\frac{\angle[\boldsymbol{A}'_q]_{w,r}}{2\pi q} = \begin{cases} [\boldsymbol{A}]_{w,r}, & \text{if } w, r \neq u_0 \\ \frac{1}{2} \cdot \text{①}, & \text{if } w \neq u_0, r = u_0 \\ \frac{1}{2} \cdot \text{②}, & \text{if } w = u_0, r \neq u_0 \end{cases}, \tag{7}$$

where

$$\text{①} = \frac{1}{2}|[\boldsymbol{A}_q]_{w,u_0} \exp\{-i\theta\}| + \frac{1}{2}\frac{\angle[\boldsymbol{A}'_q]_{w,u_0} \exp\{-i\theta\}}{2\pi q} = \boldsymbol{A}_{w,u_0} - \frac{\theta}{4\pi q} = -\frac{\theta}{4\pi q}, \tag{8}$$

and

$$\text{②} = \frac{1}{2}|[\boldsymbol{A}_q]_{u_0,r} \exp\{-i\theta\}| + \frac{1}{2}\frac{\angle[\boldsymbol{A}'_q]_{u_0,r} \exp\{-i\theta\}}{2\pi q} = \boldsymbol{A}_{u_0,r} + \frac{\theta}{4\pi q} = 1 + \frac{\theta}{4\pi q}. \tag{9}$$

Now we can see that for the fixed $u, v$, we have

$$\Phi_{u,v}(2,2) - \Phi'_{u,v}(2,2) = \boldsymbol{A}^2_{u,v} - \boldsymbol{A}'^2_{u,v} = \sum_w \boldsymbol{A}_{u,w}\boldsymbol{A}_{w,v} - \boldsymbol{A}'_{u,w}\boldsymbol{A}'_{w,v}$$

$$= \boldsymbol{A}_{u,u_0}\boldsymbol{A}_{u_0,v} - \boldsymbol{A}'_{u,u_0}\boldsymbol{A}'_{u_0,v} = 0 \cdot 1 - \frac{1}{4}\text{①} \cdot \text{②} \tag{10}$$

$$= -\frac{1}{4}\left(1 + \frac{\theta}{4\pi q}\right)\frac{\theta}{4\pi q}.$$

Now let $\theta = 4\pi q$. Then $\Phi_{u,v}(2,2) - \Phi'_{u,v}(2,2) = -1/2 \neq 0$. So the walk profile $\Phi_{u,v}(2,2), \Phi'_{u,v}(2,2)$ are different while the PE $(\lambda, z_u, z_v) = (\lambda', z'_u, z'_v)$ are the same[1]. It is

---

[1] Note that here PE denotes the eigenvalues/vectors of $\boldsymbol{A}_q$ rather than $\boldsymbol{L}_q$, as the walk profile of $\boldsymbol{A}$ is more naturally correlated to the spectrum of $\boldsymbol{A}_q$.

worthy noticing that using the same construction except that let $\boldsymbol{A}_{u_0,u_1} = 0$, we can show that $spd(u_0, u_1) = 0$ but $spd'(u_0, u_1) = 2$. Therefore, Mag-PE cannot express shortest path distance as well.

$\square$

## A.2 PROOF OF THEOREM 4.2

**Theorem 4.2.** *Let $L$ be a positive integer and let $Q = \lceil \frac{L}{2} \rceil + 1$, where $\lceil \cdot \rceil$ means ceiling. If we let $\vec{q} = (q_1, q_2, ..., q_Q)$ with $Q$ distinct $q$'s and $q_1, ..., q_{L+1} \in [0, \frac{1}{4})$, then for all $\ell \leq L$ and $k \leq \ell$, walk profile $\Phi_{u,v}(\ell, k)$ can be exactly computed from $(\boldsymbol{\lambda}^{\vec{q}}, z_u^{\vec{q}}, z_v^{\vec{q}})$, where $\boldsymbol{\lambda}^{\vec{q}}, z^{\vec{q}}$ are concatenation of eigenvalues/eigenvectors of different $q$ from $\vec{q}$.*

*Proof.* The proof starts with identifying a key relation between $[\boldsymbol{A}_q]_{u,v}^{\ell}$ and $\Phi_{u,v}^{\ell}$. Note that $[\boldsymbol{A}_q]_{u,v}^{\ell}$ equals to the sum of weight of all length-$\ell$ bidirectional walks:

$$
\begin{aligned}
[\boldsymbol{A}_q]_{u,v}^{\ell} &= \sum_{w_1, w_2, ..., w_{\ell-1}} [\boldsymbol{A}_q]_{u,w_1} [\boldsymbol{A}_q]_{w_1,w_2} ... [\boldsymbol{A}_q]_{w_\ell, v} \\
&= \sum_{\substack{\text{bi-walk } (u, w_1, ..., w_{\ell-1}, v) \\ \text{of length } \ell}} [\boldsymbol{A}_q]_{u,w_1} [\boldsymbol{A}_q]_{w_1,w_2} ... [\boldsymbol{A}_q]_{w_\ell, v}.
\end{aligned}
\tag{11}
$$

It is bidirectional because $\boldsymbol{A}_q$ is Hermitian and allows transition along the forward edges with weight $\boldsymbol{A}_{u,v} \exp\{i2\pi q\}$ or backward edges with weight $\boldsymbol{A}_{u,v}^* \exp\{-i2\pi q\}$. Note that we can categorize the bidirectional walks by their number of forward and reverse edges. Then for bidirectional walk of length $\ell$ and exact $k$ forward edges, the forward edges will cause an additional phase term $\exp\{i2\pi k\}$ while the remaining $\ell - k$ backward edges will cause $\exp\{-i2\pi(\ell-k)\} = \exp\{i2\pi(k-\ell)\}$. So there is a common phase term $\exp\{i2\pi(2k - \ell)\}$ for all bidirectional walks of length $\ell$ and $k$ forward edges. Assume there is no multi-edges, i.e., directed edges $(u, v), (v, u)$ cannot appear simultaneously, now we can re-organize the formula into

$$
\begin{aligned}
[\boldsymbol{A}_q]_{u,v}^{\ell} &= \sum_{k=0}^{\ell} \sum_{\substack{\text{bi-walk}(u, w_1, ..., w_{\ell-1}, v) \\ \text{of length } \ell \text{ and } k \text{ forward edges}}} [\boldsymbol{A}_q]_{u,w_1} [\boldsymbol{A}_q]_{w_1,w_2} ... [\boldsymbol{A}_q]_{w_\ell, v} \\
&= \sum_{k=0}^{\ell} \sum_{\substack{\text{bi-walk}(u, w_1, ..., w_{\ell-1}, v) \\ \text{of length } \ell \text{ and } k \text{ forward edges}}} \exp\{i2\pi q(2k - \ell)\} [\boldsymbol{A}_1]_{u,w_1} [\boldsymbol{A}_2]_{w_1,w_2} ... [\boldsymbol{A}_\ell]_{w_\ell, v},
\end{aligned}
\tag{12}
$$

where $\boldsymbol{A}_1, ..., \boldsymbol{A}_\ell$ is either $\boldsymbol{A}$ or $\boldsymbol{A}^\dagger$, depending on it is a forward or backward edge. Now by taking phase term $\exp\{i2\pi q(2k - \ell)\}$ out the inner sum, we get exactly walk profile:

$$
\begin{aligned}
[\boldsymbol{A}_q]_{u,v}^{\ell} &= \sum_{k=0}^{\ell} \exp\{i2\pi q(2k - \ell)\} \sum_{\substack{\text{bi-walk}(u, w_1, ..., w_{\ell-1}, v) \\ \text{of length } \ell \text{ and } k \text{ forward edges}}} [\boldsymbol{A}_1]_{u,w_1} [\boldsymbol{A}_2]_{w_1,w_2} ... [\boldsymbol{A}_\ell]_{w_\ell, v} \\
&= \sum_{k=0}^{\ell} \exp\{i2\pi q(2k - \ell)\} \Phi_{u,v}(\ell, k) \\
&= \exp\{-i2\pi q\ell\} \sum_{k=0}^{\ell} \exp\{i4\pi qk\} \Phi_{u,v}(\ell, k).
\end{aligned}
\tag{13}
$$

Let $L$ be any positive integer. Let us write the relation above for $\Phi_{u,v}(\ell, \cdot)$ $(\ell \leq L)$ in a matrix form:

$$
\boldsymbol{F}_q \begin{pmatrix} \Phi_{u,v}(1,0) & \Phi_{u,v}(2,0) & ... & \Phi_{u,v}(L,0) \\ \Phi_{u,v}(1,1) & \Phi_{u,v}(2,1) & ... & \Phi_{u,v}(L,1) \\ 0 & \Phi_{u,v}(2,2) & ... & \Phi_{u,v}(L,2) \\ 0 & 0 & ... & \Phi_{u,v}(L,3) \\ ... & ... & ... & \Phi_{u,v}(L,L) \end{pmatrix} = \begin{pmatrix} e^{i2\pi} [\boldsymbol{A}_q]_{u,v} & e^{i4\pi} [\boldsymbol{A}_q^2]_{u,v} & ... & e^{i2L\pi} [\boldsymbol{A}_q^L]_{u,v} \end{pmatrix},
\tag{14}
$$

where $\boldsymbol{F}_q = (1, \exp\{i4\pi q\}, \exp\{i8\pi q\}, ..., \exp\{i8L\pi q\}) \in \mathbb{R}^{1 \times (L+1)}$. For notation convenience, let us denote RHS as $\boldsymbol{Y}_q$ so we write the linear system as $\boldsymbol{F}_q \boldsymbol{\Phi} = \boldsymbol{Y}_q$. Note that as $\Phi$ are real values, we can know the values of $\boldsymbol{Y}_{\frac{1}{2}-q}$ by the complex conjugate of $\boldsymbol{Y}_q$ thanks to the symmetry of Fourier matrix $\boldsymbol{F}_q$:

$$\boldsymbol{Y}_{\frac{1}{2}-q} = \boldsymbol{F}_{\frac{1}{2}-q} \boldsymbol{\Phi} = \boldsymbol{F}_q^* \boldsymbol{\Phi} = \boldsymbol{Y}_q^*. \tag{15}$$

Now we are ready to prove the theorem. Let $Q = \lceil L/2 \rceil + 1$ and let $\vec{q} = (q_1, q_2, ..., q_Q)$ with each $q_j \leq 1/4$ and $q_j$ are distinct. From eigenvalues/vectors at these frequencies we are able to get $\boldsymbol{Y}_{\vec{q}}$. Furthermore, we get to know $\boldsymbol{Y}_{\frac{1}{2}-\vec{q}} = \boldsymbol{Y}_{\vec{q}}^*$ as we just argued. That is, we know $\boldsymbol{Y}_{\vec{q}'}$ where $\vec{q}' = (\vec{q}, 1/2 - \vec{q})$. Since $\vec{q} < 1/4$, we claim that $1/2 - \vec{q} > 1/4$ and thus $\vec{q}'$ contains $Q' := 2 \cdot (\lceil L/2 \rceil + 1) \geq L + 2$ different qs. We can put equation $\boldsymbol{F}_q \boldsymbol{\Phi} = \boldsymbol{Y}_q$ for each $q$ in $\vec{q}'$ into one matrix equation as follows:

$$\boldsymbol{F}_{\vec{q}'} \begin{pmatrix} \Phi_{u,v}(1,0) & \Phi_{u,v}(2,0) & ... & \Phi_{u,v}(L,0) \\ \Phi_{u,v}(1,1) & \Phi_{u,v}(2,1) & ... & \Phi_{u,v}(L,1) \\ 0 & \Phi_{u,v}(2,2) & ... & \Phi_{u,v}(L,2) \\ 0 & 0 & ... & \Phi_{u,v}(L,3) \\ ... & ... & ... & \Phi_{u,v}(L,L) \end{pmatrix} = \begin{pmatrix} e^{i2\pi}[\boldsymbol{A}_{q_1}]_{u,v} & ... & e^{i2L\pi}[\boldsymbol{A}_{q_1}^L]_{u,v} \\ ... & ... & ... \\ e^{i2\pi}[\boldsymbol{A}_{q_{Q'}}]_{u,v} & ... & e^{i2L\pi}[\boldsymbol{A}_{q_{Q'}}^L]_{u,v} \end{pmatrix},$$
$$\tag{16}$$

where $\boldsymbol{F}_{\vec{q}} = (\boldsymbol{F}_{q_1}; \boldsymbol{F}_{q_2}; ...; \boldsymbol{F}_{q_{L+1}}) \in \mathbb{R}^{Q' \times (L+1)}$. Since $\boldsymbol{F}_{\vec{q}'}$ contains $\boldsymbol{Q}' \geq L + 2$ different frequencies and thus is full-rank, we can uniquely determine walk profile $\Phi_{u,v}$ from the RHS matrix. Notably, $q_j = \frac{j}{2(L+1)}$ makes $\boldsymbol{F}_{\vec{q}}$ **discrete Fourier transform**, a unitary matrix which is invertible by taking its conjugate transpose.

$\square$

# B  EXPERIMENTAL SETUP

In this section, we give further implementation details. We use Quadro RTX 6000 on Linux system to train the models. The training time for single run is typically between 1 hour to 5 hours.

Note that for SignNet, we use $\phi$ and $\rho$ to represent $\text{SignNet}(v_1, ..., v_d) = \rho([\phi(v_j) + \phi(-v_j)]_{j=1,...,d})$, where $v_j$ is the $i$-th eigenvectors.

For SPE, we use both node SPE $z_{\text{node}}$ and edge SPE $z_{\text{edge}}$ by default, except for High-level synthetic task where we find using $z_{\text{node}}$ only has comparable performance to both $z_{\text{node}}$ and $z_{\text{edge}}$.

For each experiment, the search space for baseline single $q$ is exactly the range of multiple $\vec{q}$. For example, if for multiple $\vec{q}$ we choose $\vec{q} = (1/10, 2/10, ..., 5/10)$, then for baseline single $q$ we search over $q = 1/10, 2/10, ..., 5/10$.

## B.1  DISTANCE PREDICTION ON DIRECTED GRAPHS

| Targets | walk profile (regular graphs) | spd (regular graphs) | lpd (regular graphs) |
|---|---|---|---|
| Base model | | 8-layer MLPs | |
| Hidden dim | | 64 | |
| Batch size | | 512 | |
| Learning rate | | 1e-3 | |
| Dropout | | 0 | |
| Epoch | | 150 | |
| Optimizer | | Adam ($\beta_1 = 0.9, \beta_2 = 0.999$) | |
| PE dim | | 32 | |
| Multiple q | (1/10, 2/10, ..., 5/10) | (1/30, 2/30, ...., 15/30) | (1/30, 2/30, ...., 15/30) |
| PE processing | SignNet ($\phi$ =3-layer MLPs, $\rho$=3-layer MLPs), SPE ($\phi$=3-layer MLPs, $\rho$=2-layer GIN) | | |

Table 4: Hyperparameter for walk profile/shortest path distance/longest path distance prediction on regular directed graphs.

| Targets | walk profile (DAG) | spd (DAG) | lpd (DAG) |
|---|---|---|---|
| Base model | 8-layer MLPs | | |
| Hidden dim | 64 | | |
| Batch size | 512 | | |
| Learning rate | 1e-3 | | |
| Dropout | 0 | | |
| Epoch | 150 | | |
| Optimizer | Adam ($\beta_1 = 0.9, \beta_2 = 0.999$) | | |
| PE dim | 32 | | |
| Multiple q | (1/10, 2/10, ..., 5/10) | (1/20, 2/20, ...., 10/20) | (1/20, 2/20, ...., 10/20) |
| PE processing | SignNet ($\phi$ =3-layer MLPs, $\rho$=3-layer MLPs), SPE ($\phi$=3-layer MLPs, $\rho$=2-layer GIN) | | |

Table 5: Hyperparameter for walk profile/shortest path distance/longest path distance prediction on directed acyclic graphs.

### B.1.1 MODEL HYPERPARAMETER

See Table 4 and 5.

### B.2 SORTING NETWORKS

| Targets | Sorting Network |
|---|---|
| Base model | 3-layer Transformer+mean pooling+3-layer MLPs |
| Hidden dim | 256 |
| Batch size | 48 |
| Learning rate | 1e-4 |
| Dropout | 0.2 |
| Epoch | 15 (SignNet) or 5 (SPE) |
| Optimizer | Adam ($\beta_1 = 0.7, \beta_2 = 0.9$, weight decay $6 \times 10^{-5}$) |
| PE dim | 25 |
| Multiple q | (1/20, 2/10, ..., 5/20) / $d_G$ |
| PE processing | SignNet ($\phi$ =3-layer MLPs, $\rho$=3-layer MLPs), SPE ($\phi$=3-layer MLPs, $\rho$=2-layer GIN) |

Table 6: Hyperparameter for sorting network prediction.

### B.2.1 MODEL HYPERPARAMETER

See Table 6.

### B.2.2 DATASET GENERATION

Our sorting networks generation follows exactly as Geisler et al. (2023). To generate a sorting network, the length of sequence (number of variables to sort) is first randomly chosen. Given the sequence length, each step it will generate a pair-wise sorting operator between two random variables, until it becomes a valid sorting network (can correctly sort arbitrary input sequence) or reaches the maximal number of sorting operators. The resulting sorting network is then translated into a directed graph, where each node represents a sorting operator, whose feature is the ids of two variables to sort. In the sorting network, if two sorting operators share a common variable to sort, the corresponding nodes will be connected by a directed edge (from the first operator to the second one).

For each generated sorting network, the test dataset further contains the reversion version of the graph, by reversing every directed edge in the directed graph. The resulting reverse sorting network is very likely not a valid sorting network.

### B.3 OPEN CIRCUIT BENCHMARK

### B.4 BI-LEVEL GNN

Note that graphs in the dataset are two-level: some directed edges describe the connection of nodes (regular edges), while some extra edges represent subgraph patterns in the graph (subgraph edges). In

| Targets | Gain | BW | PM |
|---|---|---|---|
| Base model | 4-layer bidi. GIN | 3-layer bidi. GIN | 4-layer bidi. GIN |
| Graph Pooling | Sum | Sum | Sum |
| Hidden dim | 96 | 192 | 288 |
| Batch size | 128 | 64 | 64 |
| Learning rate | 0.0067 | 0.0065 | 0.0021 |
| Dropout | 0.1 | 0 | 0.2 |
| Epoch | 300 | 300 | 300 |
| Optimizer | Adam ($\beta_1 = 0.9, \beta_2 = 0.999$) | | |
| PE dim | 10 | | |
| Multiple q | (1/100, 2/100, ..., 10/100) | (1/100, 2/100, ..., 5/100) | (1/100, 2/100, ..., 5/100) |
| PE processing | SignNet ($\phi$=1-layer bidi. GIN, $\rho$=3-layer MLPs), SPE ($\phi$=3-layer MLPs, $\rho$=2-layer bidi. GIN) | | |

Table 7: Hyperparameter for bidirected GIN on Open Circuit Benchmark.

| Targets | Gain | BW | PM |
|---|---|---|---|
| Base model | 4-layer undi. GIN | 3-layer undi. GIN | 4-layer undi. GIN |
| Graph Pooling | Sum | Sum | Sum |
| Hidden dim | 96 | 192 | 288 |
| Batch size | 128 | 64 | 64 |
| Learning rate | 0.0067 | 0.0065 | 0.0021 |
| Dropout | 0.1 | 0 | 0.2 |
| Epoch | 300 | 300 | 300 |
| Optimizer | Adam ($\beta_1 = 0.9, \beta_2 = 0.999$) | | |
| PE dim | 10 | | |
| Multiple q | (1/100, 2/100, ..., 10/100) | (1/100, 2/100, ..., 5/100) | (1/100, 2/100, ..., 5/100) |
| PE processing | SignNet ($\phi$=1-layer undi. GIN, $\rho$=3-layer MLPs), SPE ($\phi$=3-layer MLPs, $\rho$=2-layer undi. GIN) | | |

Table 8: Hyperparameter for undirected GIN on Open Circuit Benchmark.

| Targets | Gain | BW | PM |
|---|---|---|---|
| Base model | 3-layer SAT (2-hop bidi. GIN) | 3-layer SAT (2-hop bidi. GIN) | 3-layer SAT (1-hop bidi. GIN) |
| Graph Pooling | Sum | Sum | Sum |
| Hidden dim | 54 | 78 | 54 |
| Batch size | 256 | 256 | 64 |
| Learning rate | 0.004 | 0.00119 | 0.00117 |
| Dropout | 0.2 | 0.3 | 0.2 |
| Epoch | 200 | 200 | 200 |
| Optimizer | Adam ($\beta_1 = 0.9, \beta_2 = 0.999$) | | |
| PE dim | 10 | | |
| Multiple q | (1/100, 2/100, ..., 5/100) | (1/100, 2/100, ..., 10/100) | (1/100, 2/100, ..., 5/100) |
| PE processing | SignNet ($\phi$=1-layer bidi. GIN, $\rho$=3-layer MLPs), SPE ($\phi$=3-layer MLPs, $\rho$=2-layer bidi. GIN) | | |

Table 9: Hyperparameter for SAT (each layer uses bidirected GIN as kernel) on Open Circuit Benchmark.

| Targets | Gain | BW | PM |
|---|---|---|---|
| Base model | 3-layer SAT (1-hop undi. GIN) | 3-layer SAT (1-hop bidi. GIN) | 3-layer SAT (1-hop bidi. GIN) |
| Graph Pooling | Sum | Sum | Sum |
| Hidden dim | 54 | 54 | 54 |
| Batch size | 64 | 256 | 64 |
| Learning rate | 0.009 | 0.002 | 0.00117 |
| Dropout | 0.2 | 0.3 | 0.1 |
| Epoch | 200 | 200 | 200 |
| Optimizer | Adam ($\beta_1 = 0.9, \beta_2 = 0.999$) | | |
| PE dim | 10 | | |
| Multiple q | (1/100, 2/100, ..., 10/100) | (1/100, 2/100, ..., 10/100) | (1/100, 2/100, ..., 10/100) |
| PE processing | SignNet ($\phi$=1-layer undi. GIN, $\rho$=3-layer MLPs), SPE ($\phi$=3-layer MLPs, $\rho$=2-layer undi. GIN) | | |

Table 10: Hyperparameter for SAT (each layer uses undirected GIN as kernel) on Open Circuit Benchmark.

our experiment, we uniform apply a GNN to the nodes with subgraph edges only, and then apply other GNN to the nodes with regular edges.

| Targets | DSP | LUT |
|---|---|---|
| Base model | 4-layer bidi. GIN | 4-layer bidi. GIN |
| Graph Pooling | Mean | Mean |
| Hidden dim | 84 | 192 |
| Batch size | 64 | 56 |
| Learning rate | 0.003 | 0.0023 |
| Dropout | 0.2 | 0.1 |
| Epoch | 400 | 400 |
| Optimizer | Adam ($\beta_1 = 0.9, \beta_2 = 0.999$) | |
| PE dim | 10 | |
| Multiple q | (1/100, 2/100, ..., 5/100) | (1/100, 2/100, ..., 5/100) |
| PE processing | SignNet ($\phi$=2-layer bidi. GIN, $\rho$=3-layer MLPs), SPE ($\phi$=3-layer MLPs, $\rho$=2-layer bidi. GIN) | |

Table 11: Hyperparameter for bidirected GIN on High-level Synthetic dataset.

| Targets | DSP | LUT |
|---|---|---|
| Base model | 2-layer undi. GIN | 4-layer bidi. GIN |
| Graph Pooling | Mean | Mean |
| Hidden dim | 56 | 84 |
| Batch size | 32 | 32 |
| Learning rate | 0.006 | 0.0015 |
| Dropout | 0.1 | 0.3 |
| Epoch | 400 | 400 |
| Optimizer | Adam ($\beta_1 = 0.9, \beta_2 = 0.999$) | |
| PE dim | 10 | |
| Multiple q | (1/100, 2/100, ..., 5/100) | (1/100, 2/100, ..., 5/100) |
| PE processing | SignNet ($\phi$=2-layer undi. GIN, $\rho$=3-layer MLPs), SPE ($\phi$=3-layer MLPs, $\rho$=2-layer undi. GIN) | |

Table 12: Hyperparameter for undirected GIN on High-level Synthetic dataset.

| Targets | DSP | LUT |
|---|---|---|
| Base model | 4-layer SAT (2-hop bidi. GIN) | 3-layer SAT (2-hop bidi. GIN) |
| Graph Pooling | Mean | Mean |
| Hidden dim | 78 | 62 |
| Batch size | 64 | 64 |
| Learning rate | 0.004 | 0.004 |
| Dropout | 0.1 | 0.1 |
| Epoch | 400 | 400 |
| Optimizer | Adam ($\beta_1 = 0.9, \beta_2 = 0.999$) | |
| PE dim | 10 | |
| Multiple q | (1/100, 2/100, ..., 5/100) | (1/100, 2/100) |
| PE processing | SignNet ($\phi$=2-layer bidi. GIN, $\rho$=3-layer MLPs), SPE ($\phi$=3-layer MLPs, $\rho$=2-layer bidi. GIN) | |

Table 13: Hyperparameter for SAT (each layer uses bidirected GIN as kernel) on High-level Synthetic dataset.

### B.4.1 MODEL HYPERPARAMETER

See Table 7, 8, 9, 10.

### B.5 HIGH-LEVEL SYNTHETIC

### B.5.1 MODEL HYPERPARAMETER

See Table 11, 12, 13, 14.

## C VARYING SINGLE $q$ BASELINE

This section shows the effect the varying single $q$ on the performance of Mag-PE. Horizontal axis is the value of single $q$, vertical axis is the test performance. The dashed line represents the result of Multi-q Magnetic Laplacian with fixed multi-q $\vec{q}$ as described in Section 5.1.

| Targets | DSP | LUT |
|---|---|---|
| Base model | 3-layer SAT (2-hop bidi. GIN) | 3-layer SAT (2-hop bidi. GIN) |
| Graph Pooling | Mean | Mean |
| Hidden dim | 98 | 62 |
| Batch size | 64 | 64 |
| Learning rate | 0.004 | 0.004 |
| Dropout | 0.1 | 0.1 |
| Epoch | 400 | 400 |
| Optimizer | Adam ($\beta_1 = 0.9, \beta_2 = 0.999$) | |
| PE dim | 10 | |
| Multiple q | (1/100, 2/100, ..., 5/100) | (1/100, 2/100, ..., 5/100) |
| PE processing | SignNet ($\phi$=2-layer bidi. GIN, $\rho$=3-layer MLPs), SPE ($\phi$=3-layer MLPs, $\rho$=2-layer bidi. GIN) | |

Table 14: Hyperparameter for SAT (each layer uses undirected GIN as kernel) on High-level Synthetic dataset.

As shown in Figures 5, 6, 7, 8, we can see that the multiple q performance consistently beats best performance of single q, which demonstrates the benefits of using multiple q simultaneously.

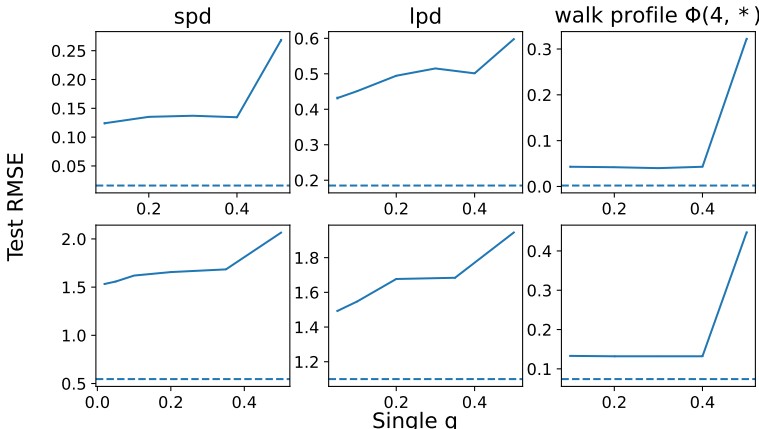

Figure 5: Test RMSE for distance prediction with varying $q$ of single-q Magnetic Laplacian. First row is for directed acyclic graphs, and second row is for regular directed graphs.

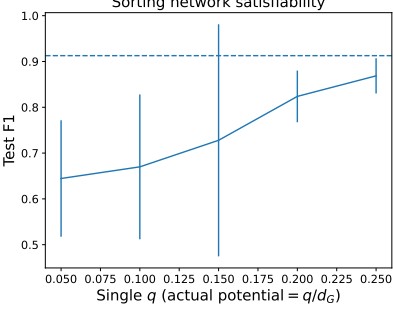

Figure 6: Test F1 of sorting network satisfibability with varying single $q$.

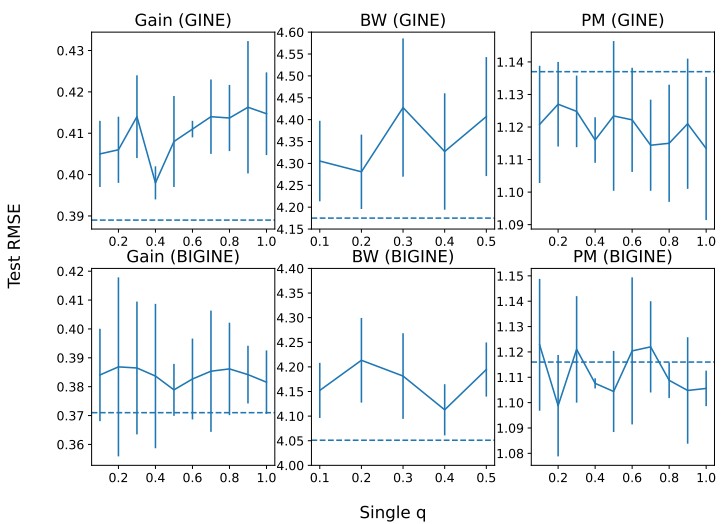

Figure 7: Test RMSE for prediction of Gain, BW, PM using backbone GINE or BIGINE, with varying $q$ of single-q Magnetic Laplacian.

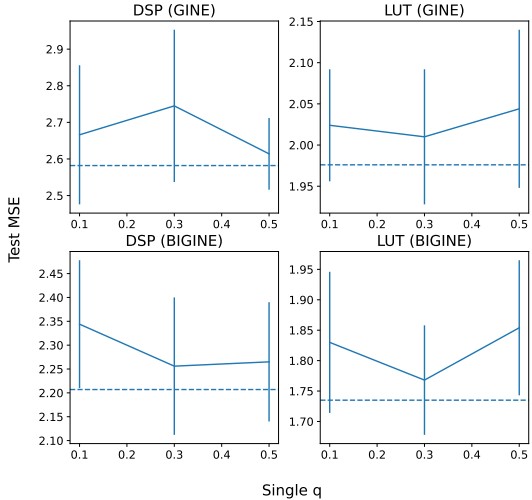

Figure 8: Test MSE for prediction of DSP and LUT, using backbone GINE or BIGINE, with varying $q$ of single-q Magnetic Laplacian.

Table 15: Test RMSE results over 3 random seeds for node-pair distance prediction.

| PE method | PE processing | Directed Acyclic Graph | | | Regular Directed Graph | | |
|---|---|---|---|---|---|---|---|
| | | $spd$ | $lpd$ | $wp(4,\cdot)$ | $spd$ | $lpd$ | $wp(4,\cdot)$ |
| Lap | Naive | $0.488_{\pm 0.005}$ | $0.727_{\pm 0.005}$ | $0.370_{\pm 0.004}$ | $2.068_{\pm 0.004}$ | $1.898_{\pm 0.001}$ | $0.480_{\pm 0.000}$ |
| | SignNet | $0.537_{\pm 0.013}$ | $0.771_{\pm 0.013}$ | $0.437_{\pm 0.000}$ | $2.064_{\pm 0.004}$ | $1.900_{\pm 0.002}$ | $0.518_{\pm 0.027}$ |
| | SPE | $0.355_{\pm 0.001}$ | $0.655_{\pm 0.002}$ | $0.326_{\pm 0.001}$ | $2.066_{\pm 0.005}$ | $1.920_{\pm 0.000}$ | $0.452_{\pm 0.001}$ |
| SVD | Naive | $0.649_{\pm 0.002}$ | $0.853_{\pm 0.002}$ | $0.721_{\pm 0.000}$ | $2.196_{\pm 0.002}$ | $1.982_{\pm 0.004}$ | $0.519_{\pm 0.000}$ |
| | SignNet | $0.673_{\pm 0.003}$ | $0.872_{\pm 0.002}$ | $0.443_{\pm 0.001}$ | $2.229_{\pm 0.003}$ | $1.996_{\pm 0.005}$ | $0.541_{\pm 0.001}$ |
| | SPE | $0.727_{\pm 0.001}$ | $0.912_{\pm 0.001}$ | $0.721_{\pm 0.000}$ | $2.261_{\pm 0.002}$ | $2.122_{\pm 0.007}$ | $0.755_{\pm 0.000}$ |
| MagLap-1q (q=0.1) | Naive | $0.366_{\pm 0.003}$ | $0.593_{\pm 0.003}$ | $0.241_{\pm 0.010}$ | $1.826_{\pm 0.005}$ | $1.760_{\pm 0.007}$ | $0.311_{\pm 0.013}$ |
| | SignNet | $0.554_{\pm 0.001}$ | $0.699_{\pm 0.002}$ | $0.268_{\pm 0.042}$ | $2.048_{\pm 0.004}$ | $1.881_{\pm 0.003}$ | $0.413_{\pm 0.001}$ |
| | SPE | $0.124_{\pm 0.002}$ | $0.433_{\pm 0.002}$ | $0.043_{\pm 0.000}$ | $1.620_{\pm 0.005}$ | $1.547_{\pm 0.004}$ | $0.133_{\pm 0.001}$ |
| MagLap-1q (best q) | SPE | $0.124_{\pm 0.002}$ | $0.432_{\pm 0.004}$ | $0.040_{\pm 0.001}$ | $1.533_{\pm 0.007}$ | $1.493_{\pm 0.003}$ | $0.132_{\pm 0.000}$ |
| MagLap-Multi-q | Naive | $0.353_{\pm 0.003}$ | $0.535_{\pm 0.006}$ | $0.188_{\pm 0.010}$ | $1.708_{\pm 0.012}$ | $1.661_{\pm 0.002}$ | $0.257_{\pm 0.007}$ |
| | SignNet | $0.473_{\pm 0.000}$ | $0.579_{\pm 0.001}$ | $0.280_{\pm 0.004}$ | $1.906_{\pm 0.001}$ | $1.784_{\pm 0.008}$ | $0.377_{\pm 0.007}$ |
| | SPE | $0.016_{\pm 0.000}$ | $0.185_{\pm 0.036}$ | $\mathbf{0.002_{\pm 0.000}}$ | $\mathbf{0.546_{\pm 0.068}}$ | $1.100_{\pm 0.007}$ | $\mathbf{0.074_{\pm 0.001}}$ |
| **MagLap-Multi-q (random $\vec{q}$)** | Naive | $0.356_{\pm 0.022}$ | $0.564_{\pm 0.040}$ | $0.204_{\pm 0.009}$ | $1.781_{\pm 0.021}$ | $1.665_{\pm 0.011}$ | $0.298_{\pm 0.033}$ |
| | SignNet | $0.469_{\pm 0.003}$ | $0.579_{\pm 0.008}$ | $0.276_{\pm 0.008}$ | $1.895_{\pm 0.013}$ | $1.769_{\pm 0.020}$ | $0.252_{\pm 0.218}$ |
| | SPE | $\mathbf{0.015_{\pm 0.002}}$ | $\mathbf{0.146_{\pm 0.010}}$ | $\mathbf{0.002_{\pm 0.000}}$ | $0.564_{\pm 0.024}$ | $\mathbf{1.095_{\pm 0.009}}$ | $\mathbf{0.074_{\pm 0.004}}$ |

| PE method | PE processing | Test F1 |
|---|---|---|
| Lap | Naive | $51.39_{\pm 2.36}$ |
| | SignNet | $49.50_{\pm 4.01}$ |
| | SPE | $56.35_{\pm 3.70}$ |
| Maglap-1q (best q) | Naive | $68.68_{\pm 9.66}$ |
| | SignNet | $76.28_{\pm 6.82}$ |
| | SPE | $86.86_{\pm 3.85}$ |
| Maglap-5q | Naive | $75.12_{\pm 12.78}$ |
| | SignNet | $72.97_{\pm 9.38}$ |
| | SPE | $\mathbf{91.27_{\pm 0.71}}$ |
| **Maglap-5q (random $\vec{q}$)** | Naive | $51.47_{\pm 19.92}$ |
| | SignNet | $50.37_{\pm 2.12}$ |
| | SPE | $89.36_{\pm 2.74}$ |

Table 16: Test F1 scores over 5 random seeds for sorting network satisfiability.

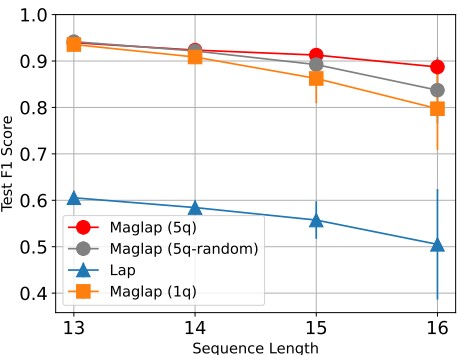

Figure 9: Test F1 scores w.r.t. different sorting network lengths.

## D RUNTIME EVALUATION

First, we note that introducing multiple q brings extra computational cost, as stated in the Conclusion and Limitations section. But we would like to emphasize that the goal of this work is to illustrate the drawbacks and benefits of single q v.s. multiple q in theory. In particular, we prove that to represent walk profiles—a bidirectional walk descriptor, multiple q are required. Thus there is a natural trade-off between complexity and expressivity. Results on prediction of various distance metrics demonstrates the superior power of multiple q, further validating our theory.

We evaluate the actual runtime of preprocessing, training and inference stage. For distance prediction, we choose connected acyclic graphs; for high-level synthetic dataset, we choose undirected GIN as backbone for evaluation. We use SPE to handle PEs and all other hyperparameters are the same as mentioned in Appendix B. We exclude Open Circuit Benchmark since the the graph size is too small to be interesting (10 to 20 nodes). Figure 4 show the runtime evaluation of multiple q for our tasks, using single Quadro RTX 6000. Preprocessing time refers to the pre-computation of eigendecompositions, and training/inference time refers to the runtime on the training set per epoch (average over 10 epochs). Note that the preprocessing time is typically not a concern, as it is far less than the total training time, e.g., on high-level synthesis total preprocessing time / total training time (400 epochs) is about 0.5%.

The dataset statistics are also included below for reference.

## E IMPACT OF MULTIPLE $q$ VALUES SELECTION POLICY

As implied by Theorem 4.2, any choice of multiple $q$ values should yields the same level of expressivity to compute walk profile. To verify this, we further implement a randomly sampled $\vec{q}$ (each $q$ is

| Dataset | Average Num. of Nodes | Average Num. of Edges |
|---|---|---|
| Distance Prediction | 27.4 | 35.8 |
| Sorting Networks | 72.8 | 272.9 |
| High-level Synthesis | 94.7 | 122.1 |

Table 17: Dataset statistics.

Table 18: Test results (RMSE for Gain/BW/PM, MSE for DSP/LUT) for circuit properties prediction. **Bold** denotes the best result for each base model, and **Bold**[†] for the best result among all base models.

| Base Model | PE method | PE processing | Gain | BW | PM | DSP | LUT |
|---|---|---|---|---|---|---|---|
| Undirected-GIN | Lap | SignNet | $0.430_{\pm 0.009}$ | $4.712_{\pm 0.134}$ | $1.127_{\pm 0.007}$ | $2.665_{\pm 0.184}$ | $2.025_{\pm 0.057}$ |
| | | SPE | $0.416_{\pm 0.021}$ | $4.321_{\pm 0.084}$ | $1.127_{\pm 0.020}$ | $2.662_{\pm 0.187}$ | $\mathbf{1.925}^{\dagger}_{\pm 0.059}$ |
| | Maglap-1q (q=0.01) | SignNet | $0.426_{\pm 0.009}$ | $4.670_{\pm 0.113}$ | $1.116_{\pm 0.009}$ | $2.673_{\pm 0.090}$ | $2.027_{\pm 0.091}$ |
| | | SPE | $0.405_{\pm 0.016}$ | $4.305_{\pm 0.092}$ | $1.121_{\pm 0.018}$ | $2.666_{\pm 0.190}$ | $2.024_{\pm 0.068}$ |
| | Maglap-1q (best q) | SPE | $0.398_{\pm 0.025}$ | $4.281_{\pm 0.085}$ | $\mathbf{1.113}_{\pm 0.022}$ | $2.614_{\pm 0.098}$ | $2.010_{\pm 0.082}$ |
| | Maglap-Multi-q | SignNet | $0.421_{\pm 0.015}$ | $4.743_{\pm 0.215}$ | $1.126_{\pm 0.011}$ | $2.665_{\pm 0.111}$ | $2.025_{\pm 0.076}$ |
| | | SPE | $\mathbf{0.389}_{\pm 0.017}$ | $\mathbf{4.175}_{\pm 0.115}$ | $1.137_{\pm 0.004}$ | $\mathbf{2.582}_{\pm 0.133}$ | $1.976_{\pm 0.089}$ |
| | **Maglap-Multi-q (random)** | SignNet | $0.409_{\pm 0.021}$ | $4.655_{\pm 0.054}$ | $1.124_{\pm 0.016}$ | $2.784_{\pm 0.157}$ | $2.087_{\pm 0.098}$ |
| | | SPE | $0.397_{\pm 0.020}$ | $4.267_{\pm 0.089}$ | $1.142_{\pm 0.011}$ | $2.772_{\pm 0.123}$ | $2.103_{\pm 0.117}$ |
| Bidirected-GIN | Lap | SignNet | $0.382_{\pm 0.008}$ | $4.371_{\pm 0.171}$ | $1.127_{\pm 0.021}$ | $2.256_{\pm 0.109}$ | $1.806_{\pm 0.096}$ |
| | | SPE | $0.391_{\pm 0.007}$ | $4.153_{\pm 0.160}$ | $1.135_{\pm 0.035}$ | $2.267_{\pm 0.126}$ | $1.786_{\pm 0.072}$ |
| | Maglap-1q (q=0.01) | SignNet | $0.388_{\pm 0.012}$ | $4.351_{\pm 0.132}$ | $1.131_{\pm 0.012}$ | $2.304_{\pm 0.143}$ | $1.882_{\pm 0.085}$ |
| | | SPE | $0.384_{\pm 0.008}$ | $4.152_{\pm 0.056}$ | $1.123_{\pm 0.026}$ | $2.344_{\pm 0.134}$ | $1.830_{\pm 0.116}$ |
| | Maglap-1q (best q) | SPE | $0.383_{\pm 0.002}$ | $4.113_{\pm 0.052}$ | $\mathbf{1.099}_{\pm 0.020}$ | $2.256_{\pm 0.144}$ | $1.768_{\pm 0.090}$ |
| | Maglap-Multi-q | SignNet | $0.381_{\pm 0.008}$ | $4.443_{\pm 0.116}$ | $1.119_{\pm 0.016}$ | $2.212_{\pm 0.116}$ | $1.791_{\pm 0.091}$ |
| | | SPE | $\mathbf{0.371}_{\pm 0.008}$ | $\mathbf{4.051}_{\pm 0.139}$ | $1.116_{\pm 0.012}$ | $\mathbf{2.207}^{\dagger}_{\pm 0.185}$ | $\mathbf{1.735}^{\dagger}_{\pm 0.096}$ |
| | **Maglap-Multi-q (random)** | SignNet | $0.378_{\pm 0.008}$ | $4.372_{\pm 0.108}$ | $1.142_{\pm 0.016}$ | $2.276_{\pm 0.170}$ | $1.823_{\pm 0.089}$ |
| | | SPE | $0.388_{\pm 0.020}$ | $4.102_{\pm 0.125}$ | $1.130_{\pm 0.018}$ | $2.499_{\pm 0.203}$ | $1.875_{\pm 0.082}$ |
| SAT (undirected-GIN) | Lap | SignNet | $0.368_{\pm 0.022}$ | $4.085_{\pm 0.189}$ | $1.038_{\pm 0.016}$ | $3.103_{\pm 0.101}$ | $2.223_{\pm 0.175}$ |
| | | SPE | $0.375_{\pm 0.016}$ | $4.180_{\pm 0.093}$ | $1.065_{\pm 0.034}$ | $3.167_{\pm 0.193}$ | $2.425_{\pm 0.168}$ |
| | Maglap-1q (q=0.01) | SignNet | $0.382_{\pm 0.009}$ | $4.143_{\pm 0.181}$ | $1.073_{\pm 0.021}$ | $3.087_{\pm 0.183}$ | $2.214_{\pm 0.150}$ |
| | | SPE | $0.366_{\pm 0.003}$ | $4.081_{\pm 0.071}$ | $1.089_{\pm 0.023}$ | $3.206_{\pm 0.197}$ | $2.362_{\pm 0.154}$ |
| | Maglap-1q (best q) | SPE | $0.361_{\pm 0.016}$ | $4.014_{\pm 0.068}$ | $1.057_{\pm 0.036}$ | $3.101_{\pm 0.176}$ | $2.362_{\pm 0.154}$ |
| | Maglap-Multi-q | SignNet | $0.368_{\pm 0.020}$ | $4.044_{\pm 0.090}$ | $1.066_{\pm 0.028}$ | $3.121_{\pm 0.143}$ | $\mathbf{2.207}_{\pm 0.113}$ |
| | | SPE | $\mathbf{0.350}^{\dagger}_{\pm 0.004}$ | $4.044_{\pm 0.153}$ | $\mathbf{1.035}^{\dagger}_{\pm 0.025}$ | $\mathbf{3.076}_{\pm 0.240}$ | $2.333_{\pm 0.147}$ |
| | **Maglap-Multi-q (random)** | SignNet | $0.361_{\pm 0.015}$ | $4.122_{\pm 0.067}$ | $1.094_{\pm 0.064}$ | $3.106_{\pm 0.123}$ | $2.236_{\pm 0.128}$ |
| | | SPE | $0.356_{\pm 0.010}$ | $\mathbf{3.953}_{\pm 0.104}$ | $1.052_{\pm 0.014}$ | $3.086_{\pm 0.174}$ | $2.320_{\pm 0.135}$ |
| SAT (bidirected-GIN) | Lap | SignNet | $0.384_{\pm 0.025}$ | $3.949_{\pm 0.125}$ | $1.069_{\pm 0.029}$ | $\mathbf{2.569}_{\pm 0.116}$ | $2.048_{\pm 0.088}$ |
| | | SPE | $0.368_{\pm 0.022}$ | $4.024_{\pm 0.106}$ | $1.046_{\pm 0.021}$ | $2.713_{\pm 0.135}$ | $2.173_{\pm 0.107}$ |
| | Maglap-1q (q=0.01) | SignNet | $0.384_{\pm 0.015}$ | $4.023_{\pm 0.032}$ | $1.055_{\pm 0.028}$ | $2.616_{\pm 0.120}$ | $2.054_{\pm 0.127}$ |
| | | SPE | $0.364_{\pm 0.012}$ | $3.996_{\pm 0.178}$ | $1.074_{\pm 0.030}$ | $2.687_{\pm 0.209}$ | $2.192_{\pm 0.135}$ |
| | Maglap-1q (best q) | SPE | $\mathbf{0.360}_{\pm 0.009}$ | $3.960_{\pm 0.060}$ | $1.062_{\pm 0.024}$ | $2.657_{\pm 0.128}$ | $2.107_{\pm 0.135}$ |
| | Maglap-Multi-q | SignNet | $0.420_{\pm 0.035}$ | $4.022_{\pm 0.128}$ | $1.089_{\pm 0.046}$ | $2.741_{\pm 0.110}$ | $2.045_{\pm 0.079}$ |
| | | SPE | $\mathbf{0.359}_{\pm 0.008}$ | $\mathbf{3.930}^{\dagger}_{\pm 0.069}$ | $1.045_{\pm 0.012}$ | $2.616_{\pm 0.151}$ | $2.082_{\pm 0.099}$ |
| | **Maglap-Multi-q (random)** | SignNet | $0.417_{\pm 0.040}$ | $4.053_{\pm 0.094}$ | $1.078_{\pm 0.043}$ | $2.677_{\pm 0.146}$ | $\mathbf{2.016}_{\pm 0.078}$ |
| | | SPE | $0.363_{\pm 0.007}$ | $3.956_{\pm 0.087}$ | $1.064_{\pm 0.022}$ | $2.690_{\pm 0.165}$ | $2.259_{\pm 0.147}$ |

uniformly sampled from $[0, 1/2]$) to predict distances and walk profile. Table 15,16,18 and Figure 9 show the results of randomly sampled multiple $q$ compared to previous methods, where evenly-spaced $\vec{q} = (\frac{1}{2L}, \frac{2}{2L}, ..., \frac{L}{2L})$ for some $L$ (see specific values in Appendix B) is the origin one we adopted in the main text, and random $\vec{q} = (q_1, q_2, ..., q_L)$ contains $L$ many $q$ values randomly sampled from $(0, 1/2)$. Note that collision $q_i \approx q_j$ could lead to a ill-conditioned recovery of walk profile, so we force that $|q_i - q_j| \geq 0.2/L$ during $q$ sampling process. For each run of model training (i.e., each random seed), we sample a different random $\vec{q}$. Table 15 shows the results. We can see that randomly sampled $\vec{q}$ has comparable and even better performance than evenly-spaced $\vec{q}$, which validates our theory.

# F   DISTANCE PREDICTION WITH GRAPH TRANSFORMERS

In Table 19, we compared the effectiveness of powerful graph transformers SAT (Chen et al., 2022) equipped with different PEs to predict various distance notions. The base graph encoder in sAT is a bidirectional GIN model. The results show that pairing previous PEs with a strong model cannot fully solve the problem of capturing graph distances, while Multi-q Mag PE still brings advantages.

# G   DISCUSSION: WALK PROFILE DIRECTLY AS FEATURES

One may wonder why not directly compute walk profile as features as alternative to multiple $q$ PE. Walk profile itself is straightforward to compute and can serve as descriptors of graph bidirectional

Table 19: Test RMSE results over 3 random seeds for node-pair distance prediction, with SAT architecture.

| Model architecture | PE method | PE processing | Directed Acyclic Graph | | |
| --- | --- | --- | --- | --- | --- |
| | | | $spd$ | $lpd$ | $wp(4, \cdot)$ |
| SAT (directed GIN) | Lap | SPE | $0.355_{\pm 0.001}$ | $0.655_{\pm 0.002}$ | $0.326_{\pm 0.001}$ |
| SAT (directed GIN) | SVD | SPE | $0.727_{\pm 0.001}$ | $0.912_{\pm 0.001}$ | $0.721_{\pm 0.000}$ |
| SAT (directed GIN) | MagLap-1q (q=0.1) | SPE | $0.124_{\pm 0.002}$ | $0.433_{\pm 0.002}$ | $0.043_{\pm 0.000}$ |
| SAT (directed GIN) | MagLap-Multi-q | SPE | $0.016_{\pm 0.000}$ | $0.185_{\pm 0.036}$ | $\mathbf{0.002_{\pm 0.000}}$ |

walk patterns, similar to random walks features on undirected graphs. However, the dimension of walk profile grows quadratically with walk length. For instance, when predicting the largest path distance on synthetic directed graphs, the longest path distance can be at most around 15, and thus we use 15 qs (we find that decreasing the number of q, e.g., to 10, will fail to fit distance greater than 10). In this case, the total dimension of the walk profile is about (16*15)/2=120. In contrast, multi-q PE (with only top K eigenvectors, K=32 in our example) usually yields a good approximation, as shown by the experimental results. Besides, we believe there are further potential advantages of multiple-q positional encodings. For instance, if there are some sparsity structures of walk profile (which is indeed observed in our experiments), it is possible to downsample a small number of q to recover the walk profile of large length. Finally, walk profiles are inherently features tied to node pairs, so it generally requires a GNN model to handle node-pair features and yields quadratic complexity. In contrast, positional encodings are node features that are more scalable, especially one may choose top K eigenvectors instead of full eigenvectors.

## H  NORMALIZED WALK PROFILE

We define normalized walk profile $\hat{\Phi}_{u,v}(\ell, k)$ by replacing every $\boldsymbol{A}$ with random walks $\boldsymbol{D}_{\text{total}}^{-1}\boldsymbol{A}$, where $[\boldsymbol{D}_{\text{total}}]_{u,u}$ is the total node degree of node $u$. As a result, $\hat{\Phi}_{u,v}(\ell, k)$ represents the probability of landing at node $v$ in a bidirectional random walk with $k$ forward edges, starting from node $u$. Note that normalized walk profiles still are able to encode important structural/distance information, such as directed shortest/longest path distance.

The reason why considering normalized walk profile is to be consistent with the definition of normalized Laplacian $\hat{\boldsymbol{L}} = \boldsymbol{I} - \boldsymbol{D}_{\text{total}}^{-1/2}\boldsymbol{A}_q\boldsymbol{D}_{\text{total}}^{-1/2}$. The normalization nature of $\hat{\boldsymbol{L}}$ makes it more suitable to recover normalized walk profile instead of the unnormalized ones.

