# OpenReview forum: "What Are Good Positional Encodings for Directed Graphs?"
_ICLR.cc/2025/Conference — ICLR 2025 Poster_

### Official Review · Reviewer_XPm4 · 2024-10-31

**Soundness:** 3
**Presentation:** 2
**Contribution:** 3
**Rating:** 5
**Confidence:** 4

**Summary:**

The paper introduces a novel positional encoding (PE) to enhance both message-passing GNNs and graph transformers, specifically, over directed graphs. Essentially, it extends the Magnetic Laplacian eigenvector-based PE by incorporating multiple potential factors. The evaluation is rather detailed containing both general performance comparisons and more elaborate analysis.

(Please note: I did not check all theoretical details)

**Strengths:**

- The research is well motivated in the first sections of the paper
- The theory is well-explained
- It is nice that the approach is considered with both MPNNs and transformers
- Also, the authors mention that this is the first work to propose the usage of complex PEs in a stable way. But I don't have the expertise to judge the relevance of this.

**Weaknesses:**

- The paper is rather densely written
- Table 3 (which contains the more important, real-world data) is hard to parse and the differences seem oftentimes marginal (e.g., considering standard deviation)
- Another suitable baseline on the DAG data would be DAG transformer [1]
- There is a recent paper [2], which considers other realistic datasets, and maybe should be considered as competitor and be compared to

[1] Luo et al. Transformers over Directed Acyclic Graphs, NeurIPS'23 https://openreview.net/pdf?id=g49s1N5nmO

[2] Wang et al. Directed Graph Transformers, TMLR'24 https://openreview.net/pdf?id=otTFPjziiK

**Questions:**

- Sec. 5.4: what is the size of the data here?

---

> ### Author Response · Authors · 2024-11-22
> **Author Response**
>
> We thank the reviewer for the constructive suggestions. Here is our response.
>
> > **Weakness 1&2**: The paper is rather densely written. Table 3 (which contains the more important, real-world data) is hard to parse and the differences seem oftentimes marginal (e.g., considering standard deviation).
>
> Thank you for the comment. To make Table 3 easier to parse, we now highlight the results that are statistically significant.
>
> > **Weakness 3**: Another suitable baseline on the DAG data would be DAG transformer [1].
>
> In our datasets, the sorting network dataset is naturally a DAG dataset, and other tasks are not necessarily DAG as we focus on general directed graphs. We implement DAGformer on sorting dataset. Note that in DAGformer, nodes only attend to other reachable nodes and it leverages the node depth to source nodes as PE.  The results are shown in the table below. We can see that Multi-q Mag-PE outperforms it for every test sequence length.
>
> | Model | Overall Test F1 | Test F1 (seq-len 13) |  Test F1 (seq-len 14) | Test F1 (seq-len 15) | Test F1 (seq-len 16) |
> | --------  | -------- | -------- | -------- | -------- | -------- |
> Transformer (Multi-q Mag-PE) | 91.27+-0.71 | 93.95+-0.20|92.35+-0.48|91.28+-0.57|88.72+-1.73
> DAGformer |77.90+-1.46 | 86.34+-0.15 | 82.52+-0.52	| 77.23+-1.68|69.01+-3.80
>
> > **Weakness 4**: There is a recent paper [2], which considers other realistic datasets, and maybe should be considered as competitor and be compared to.
>
> Thank you for introducing the paper and datasets. Unfortunately, adding a new dataset requires extensive comparison to different types of PE (Lap, SVD, Maglap) and PE methods (e.g., SignNet, SPE), which would take more time than the rebuttal period allows. We have cited this paper in the revised version and will include these experiments in the final version of this work.
>
> > **Question 1**: Sec. 5.4: what is the size of the data here?
>
> Dataset stat can be found at Table 17 in Appendix D. Here is the graph size. Distance prediction dataset: Ave.#nodes=27.4, Ave.#edges=35.8; Sorting network dataset: Ave.#nodes=72.8, Ave.#edges=272.9; High-level synthetic dataset: Ave.#nodes=94.7, Ave.#edges=122.1.

---

> ### Comment · Reviewer_XPm4 · 2024-11-25
>
> Thank you. The comparison to DAGformer is interesting. My Q1 intended to say that the numbers would help in this section to interpret the graphs. Overall, I believe my score aligns well with my impression of the paper.

---

> > ### Author Response · Authors · 2024-11-30
> >
> > Thank you so much for your kind response!

---

### Official Review · Reviewer_eJc6 · 2024-11-01

**Soundness:** 3
**Presentation:** 3
**Contribution:** 2
**Rating:** 6
**Confidence:** 3

**Summary:**

This paper addresses the need for improved positional encodings (PEs) specifically designed for directed graphs. The authors introduce Multi-q Magnetic Laplacian PE, an enhancement of the Magnetic Laplacian PE that incorporates multiple potential values (q) to better capture directed graph structure. This Multi-q Mag-PE method is proposed to address limitations in existing PEs for directed graphs by generalizing walk-counting sequences into a walk profile. The paper provides theoretical support for Multi-q Mag-PE's effectiveness and evaluates it on multiple tasks, where it is shown to outperform traditional single-q PEs and other established methods.

**Strengths:**

1.	Multi-q Mag-PE is a unique extension within the domain of GNN, specifically targeting the gap in encoding methods for directed graphs.

2.	The paper establishes theoretical backing for Multi-q Mag-PE's ability to capture bidirectional walks and to accurately express walk profiles. This is an advancement over prior methods that fall short in representing such details in directed graphs.

3.	Through extensive experiments across synthetic and real-world datasets, the paper demonstrates that Multi-q Mag-PE can outperform existing PEs, validating its effectiveness for both complex and directed graph tasks.

**Weaknesses:**

1.	Although theoretically the Multi-q Mag-PE should allow near-perfect reconstruction of the walk profile, practical RMSE results are not close to zero. This discrepancy, acknowledged by the authors, indicates limitations in applying the theoretical model to real-world scenarios.

2.	The Multi-q approach significantly increases computational demands due to multiple eigenvalue decompositions, which could challenge its scalability to larger graphs, especially those with millions of nodes or edges. Although the authors attempt to address this with runtime experiments and discuss potential optimizations, questions about the approach's feasibility for large-scale real-world applications remain.

3.	The motivation for introducing the walk profile is not so clear. To be specific, heuristic definitions or possible theoretical explanations are needed to illustrate its effectiveness.

4.	The improvements shown in real-world tasks were less pronounced than those on synthetic datasets, potentially due to the controlled nature of synthetic datasets, which may not represent the broader spectrum of real-world graph structures.

**Questions:**

1.	Given that the computational complexity of directly using the walk profile is potentially manageable, could you elaborate further on why it wasn’t directly computed as features? Additionally, would an ablation of such a direct computation provide insight into how Multi-q PE compares in practical tasks?

2.	Would it be possible to apply this model to larger graphs by selectively reducing the number of q values?

3.	Have you considered validating the proposed method on a wider range of GNN or transformer backbone models? This could help generalize the results and illustrate broader applicability across different architectures.

---

> ### Author Response · Authors · 2024-11-22
> **Author Response (1/2)**
>
> We would like to thank the reviewer for the constructive feedbacks. Here is our response.
>
> > **Weakness 1**: Although theoretically the Multi-q Mag-PE should allow near-perfect reconstruction of the walk profile, practical RMSE results are not close to zero. This discrepancy, acknowledged by the authors, indicates limitations in applying the theoretical model to real-world scenarios.
>
> Thank you for raising the concern. We think the non-exact-zero RMSE is due to optimization problems rather than expressivity  problems. We found we can directly compute ground-truth labels from multi-q PE using Eq. (2), which validates our theory. But in practice we found it could be difficult to optimize a MLP to perfectly fit the ground-truth function.
>
> > **Weakness 2**: The Multi-q approach significantly increases computational demands due to multiple eigenvalue decompositions, which could challenge its scalability to larger graphs, especially those with millions of nodes or edges. Although the authors attempt to address this with runtime experiments and discuss potential optimizations, questions about the approach's feasibility for large-scale real-world applications remain.
>
> We acknowledge that runtime is a limitation of multiple q, as discussed in Line 304 under Limitations. However, we would like to emphasize that the primary goal of this work is to examine: (1) the theoretical trade-offs of using single q versus multiple q; (2) the practical implication of multiple q. Specifically, we aim to demonstrate the increased expressive power and its empirical benefits that comes with multiple q, albeit with some computational cost.
>
> As shown in Table 1, increasing the number of q allows us to capture graph distances more effectively, illustrating the natural trade-off between expressivity and computational complexity. Multiple q also brings performance gain on sorting networks and circuit datasets as shown in Tables 2 and 3.
>
> There are several ways to mitigate the computational overhead of performing multiple eigenvalue decompositions:
> - Parallelization: The computations for multiple eigenvalue decompositions can be executed in parallel, significantly reducing runtime.
> - Efficient Algorithms: In practice, we only compute the smallest eigenvalues and their associated eigenvectors, for which efficient methods such as the Lanczos algorithm can be employed.
>
> By leveraging these optimizations, we believe the computational overhead can be alleviated, making the Multi-q approach more feasible for larger graphs.
>
> > **Weakness 3**: The motivation for introducing the walk profile is not so clear. To be specific, heuristic definitions or possible theoretical explanations are needed to illustrate its effectiveness.
>
> In Line 52 and Section 4.1, we already highlighted the importance of bidirectional walks and walk profiles in capturing key directed motifs, such as feed-forward loops. These motifs play a crucial role in many real-world applications, and walk profiles offer an effective mechanism to represent them.
>
> Regarding the request for a “theoretical explanation,” we are uncertain about the specific type of theoretical explanation the reviewer is seeking. From our perspective, the utility of walk profiles is better demonstrated by their ability to model and capture essential structural patterns relevant to practical applications, as the motif example mentioned above and the advantages of the learning models in the real-world dataset evaluation demonstrated in our experiments

---

> > ### Comment · Reviewer_eJc6 · 2024-11-22
> >
> > I appreciate the authors' responses, which have addressed my concerns to some extent.
> >
> > After reading the paper and considering all reviews and responses, I acknowledge that the work makes both theoretical and empirical contributions. However, it retains certain limitations, such as low efficiency. As a result, I will maintain my original score.

---

> > > ### Author Response · Authors · 2024-11-30
> > >
> > > Thank you so much for your kind response and confirmation of your positive support for this work!

---

> ### Author Response · Authors · 2024-11-22
> **Author Response (2/2)**
>
> > **Question 1**: Given that the computational complexity of directly using the walk profile is potentially manageable, could you elaborate further on why it wasn’t directly computed as features? Additionally, would an ablation of such a direct computation provide insight into how Multi-q PE compares in practical tasks?
>
> Thank you for the thoughtful question. In Appendix F, we discussed the advantages and limitations of directly using walk profiles as features. While using walk profiles is a viable solution, it does not diminish the utility of Multi-q PE. There are several key reasons for this:
> - Dimensionality Growth: The dimensionality of the walk profile grows quadratically with the walk length, whereas the dimensionality of Multi-q PE, which leverages only the top K eigenvectors, grows linearly. Despite this reduction, Multi-q PE still provides a good approximation to the walk profiles.
> - Node-Pair vs. Node Features: Walk profiles are inherently features tied to node pairs, which introduces additional complexity, as handling node-pair features typically results in quadratic computational complexity. In contrast, positional encodings like Multi-q PE are node-based features, enabling the use of more scalable models.
>
> Directly computing walk profiles could provide additional insights, and we agree that an ablation study comparing this approach to Multi-q PE in practical tasks would be valuable. However, since directly using walk profiles requires injecting node-pair features, incorporating them fairly would necessitate modifying the backbone models used in this work, which would take more time than the rebuttal period allows.
>
> > **Question 2**: Would it be possible to apply this model to larger graphs by selectively reducing the number of q values?
>
> We are working on a follow-up work to reduce the number of q while preserving a similar level of expressive power. Specifically, Eqn. (2) tells us solving walk profiles from Multi-q Mag-PE is a linear inverse problem. Certain priors of walk profiles can be leveraged to reduce the number of measures (number of q). For example, when walk profiles are sparsely supported, the technique of compressed sensing for Fourier measurement matrix [1,2] allows us to apply a downsample strategy to q and it still can perfectly recover walk profiles. This can significantly reduce the number of q.
>
> > **Question 3**: Have you considered validating the proposed method on a wider range of GNN or transformer backbone models? This could help generalize the results and illustrate broader applicability across different architectures.
>
> Thank you for the suggestion. In our experiments we include basic models such as GIN and vanilla transformers, as well as powerful models as SAT. We also compare the undirected version and bidirected version of these models on real-world datasets. Due to the limited time of rebuttal period, it is hard to employ new base models and compare it with all the directed PEs.
>
> [1] Donoho, David L. "Compressed sensing." IEEE Transactions on information theory 52.4 (2006): 1289-1306.
>
> [2] Vidyasagar, Mathukumalli. An introduction to compressed sensing. Society for Industrial and Applied Mathematics, 2019.

---

### Official Review · Reviewer_vWvy · 2024-11-03

**Soundness:** 2
**Presentation:** 2
**Contribution:** 2
**Rating:** 5
**Confidence:** 3

**Summary:**

This paper attempts to determine good positional encodings for directed graphs. To quantify this, the authors first introduce the graph-topological concept of “walk profilesˮ, which count how many paths there are between all two nodes with a varying relaxation $k$ of how many edges can be oriented in the “wrong wayˮ. Then, they show that a normal Laplacian PE cannot capture all of these walk profiles, while their extended Magnetic Laplacian, $L$-MagPE, can capture all such walks up to length $\frac{L}{2} +1$.

**Strengths:**

- The paper is presented adequately, and is easy to read.
- The interest in being able to express the walk profiles is nicely motivated in the introduction.
- Walk profiles are introduced as a way of counting walks which is more comprehensive than simple shortest path distances, which can be trivial in directed graphs.
- The authors show empirically that their method is superior when simple models are used on simple tasks. They also show that theoretically their PE is more expressive with respect to walk profiles.

**Weaknesses:**

- As the authors have discussed it in Section 4.3, the main limitation is the extra compute-time with larger $L$, induced namely by the calculation of large and numerous eigendecompositions. As shown in Section 5.4, the cost seems to grow linearly.
- While it is described informally why walk profiles can be of interest, there is no theoretical justification of their use. Moreover, there are no connections to existing measures of modelsʼ expressivity (e.g., $k$-WL[1]).
- It is not clear whether the authors have attempted any other methods to provide information about walk profiles. It is also not discussed whether other PEs [2, 3] also lack this ability to capture walk profiles.
- In the experiments, while there is a fair share of ablations, standard GNN models can be added.
- In the experimental section, it seems that the biggest gain was in the Experiment 5.1, where very simple models are used. As soon as the task gets more complex, a stronger model is invoked (SAT), and there the gains are much less evident if present at all.

Overall, the paper also does not adequately answer the question it sets in its title. While the paper propose a positional encoding specific for directed graphs (and discusses its properties), it does not answer the question of what makes good positional encodings for directed graphs (and where the expressiveness is located compared to many potential alternatives).

[1] Morris, C., Ritzert, M., Fey, M., Hamilton, WL., Lenssen, JE., Rattan, G., Grohe, M. “Weisfeiler and Leman Go Neural: Higher-order Graph Neural Networksˮ (2019). https://arxiv.org/abs/1810.02244

[2] Dwivedi, VP., Luu, AT., Laurent, T., Bengio, Y., Bresson, X. “Graph Neural Networks with Learnable Structural and Positional Representationsˮ (2022). https://arxiv.org/abs/2110.07875

[3] Li, P., Wang, Y., Wang, H., Leskovec, J, “Distance Encoding: Design Provably More Powerful Neural Networks for Graph Representation Learningˮ (2020). https://arxiv.org/abs/2009.00142

**Questions:**

1. Have you tried reducing the cost when calculating higher terms? Or is it a fundamental limitation of the method?

2. Why are walk profiles are of interest? Of course, they might be explicitly such for some tasks, but can graph transformers (such as SAT) not learn them some other way using other positional embeddings?

3. Have you tried other simple models on the datasets, potentially with no positional encodings just to see how they perform as baselines?

4. For the experiment in Section 5.1, have you tried larger graphs?

5. Do you know whether walk profiles are related in any way to $1$-WL or beyond for measuring expressivity?

6. How could the proposed approach be located within existing PEs in terms of expressiveness? The proposed approach may be better (as argued) in terms of capturing walk profiles, but other PEs (or their variations on directed graphs) may have other strengths.

---

> ### Author Response · Authors · 2024-11-22
> **Author Response (1/2)**
>
> We thank the reviewer for the insightful comments. Here is our response.
>
> > **Weakness 1**: Computational complexity.
>
> We acknowledge that runtime is a limitation of multiple q, as discussed in Line 304 under Limitations. However, we would like to emphasize that the primary goal of this work is to study: (1) the theoretical trade-offs of using single q v.s. multiple q; (2) the practical implication of multiple q. Specifically, we aim to demonstrate the increased expressive power and its empirical benefits that comes with multiple q, albeit with some computational cost.
>
> As shown in Table 1, increasing the number of q allows us to capture graph distances more effectively, illustrating the natural trade-off between expressivity and computational complexity. Multiple q also brings performance gain on sorting networks and circuit datasets as shown in Tables 2 and 3.
>
> There are several ways to mitigate the computational overhead of performing multiple eigenvalue decompositions:
> - Parallelization: The computations for multiple eigenvalue decompositions can be executed in parallel
> - Efficient Algorithms: In practice, we only compute the smallest eigenvalues and their associated eigenvectors, for which efficient methods such as the Lanczos algorithm can be employed.
>
> By leveraging these optimizations, we believe the computational overhead can be alleviated, making the Multi-q approach more feasible for larger graphs.
>
> > **Weakness 2** : no theoretical justification of walk profiles' use and no connections to existing measures of modelsʼ expressivity.
>
> In Line 52 and Section 4.1, we highlighted the importance of bidirectional walks and walk profiles in capturing key directed motifs, such as feed-forward loops. These motifs play a crucial role in many real-world applications, and walk profiles offer an effective mechanism to represent them.
>
> Regarding the request for a “theoretical justification,” we are uncertain about the specific type of justification the reviewer is seeking. From our perspective, the utility of walk profiles is demonstrated by their ability to model and capture essential structural patterns relevant to practical applications.
>
> To further elaborate, walk profiles are clearly more expressive than 1-WL (i.e., the subtree kernel) as illustrated by the following example of two non-isomorphic directed graphs that cannot be distinguished by 1-WL:
> $E_1=\{(1,2), (1,3), (3,2), (4,5), (4,6), (5,6)\},$
> $E_2=\{(1,2), (1,3), (5,2), (4,5), (4,6), (3,6)\}.$
>
> While we acknowledge the utility of k-WL for measuring expressivity, our work does not measure the expressive power of walk profiles in terms of k-WL [1]. This is because k-WL is primarily associated with WL kernels to measure the similarity between graphs, which differs significantly from the walk-based approaches/kernels we adopt. Furthermore, even within the current graph learning literature, k-WL is often used to evaluate models’ expressive power for **undirected** graphs. The study of expressivity metrics for directed graphs, especially metrics that sufficiently incorporate edge direction information, remains relatively underexplored. Our work perhaps is the first one that provides a direction for this, to the best of our knowledge.
>
> > **Weakness 3**: It is not clear whether the authors have attempted any other methods to provide information about walk profiles. It is also not discussed whether other PEs [2, 3] also lack this ability to capture walk profiles.
>
> In Section 4.2, we discussed why previous directed PEs cannot express walk profiles. Specifically, methods like [2, 3] were originally designed for **undirected** graphs, and their inability to capture walk profiles stems from the lack of directionality in their formulations.
> While it may be possible to generalize methods like distance encoding [3] to directed graphs, doing so is non-trivial. Incorporating direction information in such methods would require significant modifications to their original designs. As it stands, the original versions of these approaches cannot capture walk profiles due to their inherent lack of directional information.
>
> > **Weakness 4**: In the experiments, while there is a fair share of ablations, standard GNN models can be added.
>
> Thank you for your suggestion. We provide the GNNs’ results w/o PE to the real-world dataset, which can be found at Table 3 in the revised manuscript.
>
>
> [1] Morris, C., Ritzert, M., Fey, M., Hamilton, WL., Lenssen, JE., Rattan, G., Grohe, M. “Weisfeiler and Leman Go Neural: Higher-order Graph Neural Networksˮ (2019). https://arxiv.org/abs/1810.02244
>
> [2] Dwivedi, VP., Luu, AT., Laurent, T., Bengio, Y., Bresson, X. “Graph Neural Networks with Learnable Structural and Positional Representationsˮ (2022). https://arxiv.org/abs/2110.07875
>
> [3] Li, P., Wang, Y., Wang, H., Leskovec, J, “Distance Encoding: Design Provably More Powerful Neural Networks for Graph Representation Learningˮ (2020). https://arxiv.org/abs/2009.00142

---

> ### Author Response · Authors · 2024-11-22
> **Author Response (2/2)**
>
> > **Weakness 5**: In the experimental section, it seems that the biggest gain was in the Experiment 5.1, where very simple models are used. As soon as the task gets more complex, a stronger model is invoked (SAT), and there the gains are much less evident if present at all.
>
> To study whether a powerful model like SAT can surpass the merits of Multi-q,  we implement the SAT as the base model (a bidirectional GIN is used for SAT’s local graph encoder) on Experiment 5.1, distance prediction on directed acyclic graphs. The results are shown in the table below. Though overall better than the results of simple base models like MLP, we can see that existing PE even paired with SAT still cannot express graph distances well. In contrast, SAT+Multi-q Mag-PE’s test errors are 3-10 times lower than SAT+other PE. Particularly,SAT+Multi-q Mag-PE has a relatively very low error of predicting longest path distance compared. This means pairing previous PEs with a strong model cannot fully solve the problem of capturing graph distances.
>
>
> | Model |PE | Shortest Path Distance | Longest Path Distance | Walk Profile $wp(4,*)$
> | --------  | -------- | -------- | -------- |  -------- |
> SAT (BIGINE) | Lap |0.158+-0.005 |0.268+-0.002|0.080+-0.001|
> SAT (BIGINE) | SVD | 0.295+-0.002|0.353+-0.005|0.276+-0.005|
> SAT (BIGINE) | Maglap (1q) |0.060+-0.002|0.152+-0.002|0.025+-0.000|
> SAT (BIGINE) | Maglap (Multi-q) | 0.018+-0.006|0.037+-0.002|0.002+-0.000|
>
>
> > **Question 1**: Have you tried reducing the cost when calculating higher terms? Or is it a fundamental limitation of the method?
>
> We are working on a follow-up work to reduce the number of q while preserving a similar level of expressive power. Specifically, Eqn. (2) tells us solving walk profiles from Multi-q Mag-PE is a linear inverse problem. Certain priors of walk profiles can be leveraged to reduce the number of measures (number of q). For example, when walk profiles are sparsely supported, the technique of compressed sensing for Fourier measurement matrix [5,6] allows us to apply a downsample strategy to q and it still can perfectly recover walk profiles. This can significantly reduce the number of q.
>
> > **Question 4**: For the experiment in Section 5.1, have you tried larger graphs?
>
> We adapted the same dataset config from Geisler [4], for which the graph size ranges from 16 to 83.
>
> > **Question 2,3,5,6**
>
> Please kindly see our response to weaknesses above .
>
>
> [4] Geisler, Simon, et al. "Transformers meet directed graphs." International Conference on Machine Learning. PMLR, 2023.
>
> [5] Donoho, David L. "Compressed sensing." IEEE Transactions on information theory 52.4 (2006): 1289-1306.
>
> [6] Vidyasagar, Mathukumalli. An introduction to compressed sensing. Society for Industrial and Applied Mathematics, 2019.

---

> > ### Author Response · Authors · 2024-11-30
> >
> > Dear Reviewer, we were wondering if the response above has addressed your concerns. As it has been some time since we left these comments and the discussion period is approaching its close, we would greatly appreciate your feedback.
> >
> > Thank you in advance!

---

### Official Review · Reviewer_BH1v · 2024-11-03

**Soundness:** 4
**Presentation:** 4
**Contribution:** 4
**Rating:** 8
**Confidence:** 4

**Summary:**

This paper proposes a new positional encoding for node representation in directed graphs with provably better expressive power and stability. The paper starts by discussing Magnetic Laplacians with potential $q$, and introduces the concept of *walk profiles*, the number of walks of lengths $l$ given $l - k$ backwards passes of an edge, a more informative descriptor of directed graphs than the standard adjacency powers could provide. It then shows how a Magnetic Laplacian fails to predict walk profiles. Given this, the paper then proposes a simple extension, Multi-q Magnetic Laplacian Positional Encodings, and shows that this extension, with $Q$ distinct q values selected from the range $[0, \frac{1}{4})$ can predict walk profiles of length $L$ of nearly double $Q$ (Precisely, $Q = \lceil \frac{L}{2}\rceil + 1$). The paper then studies the stability of complex positional encodings returned with multiple Qs and proposes an invariant encoding mechanism (SPE) to circumvent the increasing ambiguity of complex eigenvectors and eigenvalues.

Empirically, the paper conducts extensive experimentation, both on real-world and synthetic baselines. In synthetic settings, the paper proposes a setting in which models must predict (normalized) walk profiles and other structural properties (shortest path, longest path) over directed graphs. In these experiments, Multi-q Magnetic Laplacian PE with SPE clearly demonstrates superior performance, even with respect to the same model with other encoding choices than SPE (such as SignNet). Moreover, the paper reports ablation results on the selection of $q$ values, showing that multiple $q$ values are indeed necessary for performance, and that even only using the best $q$ cannot reproduce the best results. A robustness study is also reported, highlighting that performance is robust to choices of $q$, and even improves slightly with random (distinct) $q$ values. In terms of real-world experiments, the paper conducts an extensive experimental study on a suite of directed graph benchmarks, demonstrating strong performance improvements with Multi-q Magnetic Laplacian PE across several baseline models. Finally, the paper discusses runtime for pre-processing, training and inference using their approach, and shows that while $q$ does lead to increased runtimes, these increases are palatable.

**Strengths:**

- The theoretical arguments made by the paper are sound and well-thought: The definition and accompanying figures for walk profiles are very helpful, and are very well connected to real-world settings. Moreover, the proof sketch for showing how Multi-q Magnetic Laplacian can recover walk profiles in Theorem 4.2 appears sound. Overall, the flow of argumentation in this paper is smooth, balanced, and easy to follow, with all pre-requisite results presented in good time.
- The experimental analysis is compelling and extensive. I especially appreciate the synthetic results and the targeted study of structural graph property predictions, which in my opinion really emphasizes the contribution of the paper's approach. These findings are also further supported by ablation and robustness analysis, and comprehensive details in the appendix. Real-world experiments are also strong and convincing. All in all, the empirical analysis ties very nicely with the theoretical results, and corroborates its results nicely.

**Weaknesses:**

- One aspect of this work that could be improved is to include baselines and datasets used in previous work, such as OGBG-code, NA and BN (as used, e.g., in Thost et al). The current results are quite strong, but I imagine that the message would be even more compelling if your findings also apply to more specialized models, and produce results on the corresponding datasets to glean more interesting insights, e.g., can these PEs even improve specialized directed GNN models?

In general, I find that this is a serious work with good presentation, extensive experimentation, and a strong take-away message. I therefore support this paper for publication.

**Questions:**

No questions.

---

> ### Author Response · Authors · 2024-11-22
> **Author Response**
>
> We sincerely thank the reviewer for the positive comments.
>
> > **Weakness 1**: One aspect of this work that could be improved is to include baselines and datasets used in previous work, such as OGBG-code, NA and BN (as used, e.g., in Thost et al). The current results are quite strong, but I imagine that the message would be even more compelling if your findings also apply to more specialized models, and produce results on the corresponding datasets to glean more interesting insights, e.g., can these PEs even improve specialized directed GNN models?
>
> Thank you for introducing the datasets. Unfortunately, adding a new dataset requires extensive comparison to different PE (Lap, SVD, Maglap) and PE methods (SignNet, SPE), which would take more time than the rebuttal period allows.

---

> > ### Comment · Reviewer_BH1v · 2024-11-26
> > **Reviewer Response**
> >
> > Thank you for your response! I maintain my rating.

---

> > > ### Author Response · Authors · 2024-11-30
> > >
> > > Thank you so much for your kind response and confirmation of your positive support for this work!

---

### Official Review · Reviewer_EKAQ · 2024-11-04

**Soundness:** 3
**Presentation:** 4
**Contribution:** 3
**Rating:** 8
**Confidence:** 4

**Summary:**

The authors propose (i) the notion of walk profile in a directed graph for counting bidirectional walks with given numbers of forward and backward edges between two nodes and (ii) a novel positional encoding (PE) which can provably reconstruct the walk profiles of the directed graph. This PE is a direct extension of Magnetic Laplacian PE (Mag-PE) to multiple phase shift parameters q (potentials): the position of a graph node is represented by concatenating Mag-PEs for the different parameters.

In particular, they prove that in order to reconstruct walk profiles of some length L between any two nodes, just over half of this length is the number of potentials needed in their Multi-q Mag-PE. They also identify that expressing walk profiles is robust to the choice of the q values. Additionally, they generalize their PE, using ideas from previous stable PE frameworks, towards node-stable and edge-stable representations.

In the experimental section, Multi-q Mag-PE is benchmarked against Laplacian, SVD and Mag-PE in predicting node pair distances and and classifying synthetic graphs; their PE is also coupled with backbone models to predict properties of real word circuits and its robustness explored. In most cases, Multi-q Mag-PE demonstrates a consistent performance gain over comparison baselines with up to x3 execution overhead (due basically to processing/training for more q's).

**Strengths:**

- This paper has an exemplary organization with clean definitions and explanation of fundamental developments and a balanced experimental section.

- Both the notion of the walk profile and the generalization of Mag-PE to multiple q's are well motivated and simple to reason about.

- It is important that the paper provides a provable connection of Multi-q Mag-PE (which is a latent representation) to something which is more "tangible": path type counters (i.e. walk profile) (Theorem 4.2).

**Weaknesses:**

- Runtime overhead, although quantified within the x3 envelope in Section 5.4, is a disadvantage in the approach. The combination of SPE and standard Mag-PE could strike a good balance (but then the "which q value" question will be raised). Also depending on the number of q's, runtime overhead would vary. On this front, having a rule of thumb to adopt for the cardinality of q's (i.e. Q) for a given task/dataset combination, would be very useful to have.

**Questions:**

- What is the intuition behind needing the PE of another, third node (w) in addition to the SVD-PEs of u, v nodes when discussing the shortcomings of SVD-PE? Actually SVD seems to be a good choice for compactly expressing powers of special structures like (A * A^T) or its transpose; powers of A are not the only matrices to use in the presence of both forward and backward links/edges.

- Could you provide a more detailed justification of how Equation (2) is produced? (currently: "By the definition" (line 260)).

- A practical issue with Mag-PE is what q to choose. In Multi-q Mag-PE the choice of q's does not seem to be a critical factor in performance outputs (e.g. discrete Fourier transform based vs randomly sampled). Is there a justification for this empirically observed robustness? Or again, some care choosing q's should be exercised, but now on the range from which to select q's? For example for the case of random graphs q's are in (0, 1/2) range but for sorting network (satisfiability tasks) q's lie in a much "slimmer" range (defined as a function of the number nodes and edges)?

- What is the role of m in Equations (3), (4) (also consider balancing paretheses in these: minor, unbalanced "{" instead of "(" appear in them)?

---

> ### Author Response · Authors · 2024-11-22
> **Author Response (1/2)**
>
> We would like to sincerely thank the reviewer for the positive feedbacks. Here is our response.
>
> > **Weakness 1**: Runtime overhead, although quantified within the x3 envelope in Section 5.4, is a disadvantage in the approach. The combination of SPE and standard Mag-PE could strike a good balance (but then the "which q value" question will be raised). Also depending on the number of q's, runtime overhead would vary. On this front, having a rule of thumb to adopt for the cardinality of q's (i.e. Q) for a given task/dataset combination, would be very useful to have.
>
> We acknowledge that runtime is a limitation of multiple q, as discussed in Line 304 under Limitations. However, we would like to emphasize that the primary goal of this work is to examine: (1) the theoretical trade-offs of using single q versus multiple q; (2) the practical implication of multiple q. Specifically, we aim to demonstrate the increased expressive power and its empirical benefits that comes with multiple q, albeit with some computational cost.
>
> As shown in Table 1, increasing the number of q allows us to capture graph distances more effectively, illustrating the natural trade-off between expressivity and computational complexity. Multiple q also brings performance gain on sorting networks and circuit datasets as shown in Tables 2 and 3.
>
> Regarding the choice of number of q, it is generally treated as a hyperparameter in our experiment.  Empirically, we find Q=5 usually gives pretty decent results for most real-world tasks we have.
>
>
> > **Question 1**: What is the intuition behind needing the PE of another, third node (w) in addition to the SVD-PEs of u, v nodes when discussing the shortcomings of SVD-PE? Actually SVD seems to be a good choice for compactly expressing powers of special structures like (A * A^T) or its transpose; powers of A are not the only matrices to use in the presence of both forward and backward links/edges.
>
> The SVD of $A$ is equivalent to doing the eigenvalue decomposition of both $AA^T$ and $A^TA$. So SVD-based PE can express the powers like $(AA^T)^k$ and $(A^TA)^k$, which are called alternating walks. These walks are restricted to patterns in forward and backward edges alternatively, and cannot directly capture more general bidirectional patterns by inner products of PE. For example, to compute $[A^2]\_{u,v}$, it needs to access the SVD-PE of all nodes, reconstruct Adjacency matrix and do matrix multiplication. For multi-q Mag-PE, this can be computed directly via looking up the PE of $u,v$.
>
> > **Question 2**: Could you provide a more detailed justification of how Equation (2) is produced? (currently: "By the definition" (line 260)).
>
> A detailed proof can be found at Appendix A.2. Here is a short justification. $A_q^k$ represents the k-step walks on a magnetic graph (a graph whose weighted Adjacency matrix is complex and Hermitian). Such walks aggregate the phase along different paths, where each forward edge contributes to a positive phase and a backward edge contributes to a negative phase. As a result, knowing the number of forward/backward edges along paths (i.e., walk profiles), multiplying it by the correct phase difference, allows us to precisely describe $A_q^k$.
>
> > **Question 3.1**: A practical issue with Mag-PE is what q to choose. In Multi-q Mag-PE the choice of q's does not seem to be a critical factor in performance outputs (e.g. discrete Fourier transform based vs randomly sampled). Is there a justification for this empirically observed robustness?
>
> The robustness to choice of q can be justified from the linear system Eq. (2). As long as multiple q are distinct, the linear system is well-posed so we can construct the walk profile from multiple-q Mag-PE.  The discrete Fourier transform (evenly-spaced q) is a canonical choice with good numerical stability. This robustness is validated by random graph results Table 15 in Appendix.
>
> > **Question 3.2**: Or again, some care choosing q's should be exercised, but now on the range from which to select q's? For example for the case of random graphs q's are in (0, 1/2) range but for sorting network (satisfiability tasks) q's lie in a much "slimmer" range (defined as a function of the number nodes and edges)?
>
> Though the choice of q does not have impact on expressivity (Theorem 4.2), we did observe that the test performance of random q is not always the optimal on real-world dataset Table 16, 18 (but overall still better than single q), which could be due to generalization issues. So practically, we would tune the range of q, just like we need to tune the single q value for standard Mag-PE.

---

> ### Author Response · Authors · 2024-11-22
> **Author Response (2/2)**
>
> > **Question 4**: What is the role of m in Equations (3), (4) (also consider balancing paretheses in these: minor, unbalanced "{" instead of "(" appear in them)?
>
> Thank you for pointing this out. The parentheses have a typo in it and we now fix it. The role of m is just like the “hidden dimension” of the equivariant function $\phi$. Each $\phi$ maps d numbers to another d numbers in an equivariant manner (e.g., Deepsets), and in practice we may require multiple $\phi$ to get a good expressive power. Basically, a larger m means better expressivity and less stability. See [1] for a detailed discussion of the role of m in balancing expressivity and stability.
>
>
> [1] Huang, Y., Lu, W., Robinson, J., Yang, Y., Zhang, M., Jegelka, S., & Li, P. On the Stability of Expressive Positional Encodings for Graphs. In The Twelfth International Conference on Learning Representations.

---

> > ### Comment · Reviewer_EKAQ · 2024-11-27
> >
> > Thank you very much for your detailed responses and insights. I choose to maintain my positive score.

---

> > > ### Author Response · Authors · 2024-11-30
> > >
> > > Thank you so much for your kind response and confirmation of your positive support for this work!

---

### Meta-Review · Area_Chair_1gJi · 2024-12-20

**Metareview:**

This paper focuses on position encodings (basically node embeddings) for directed graphs. They introduce a notion of a 'walk profile', which captures natural structures in directed graphs, show that existing embedding approaches cannot accurately represent the walk profile, and introduce a new embedding method based on the magnetic Laplacian with multiple phase shifts that can capture the walk profile. They demonstrate that their method is effective in a variety of applications.

The reviewers found the paper to be well-written and well-motivated. Some weaknesses brought up were the computational complexity of the method and the lack of a theoretical justification behind the walk profile, outside its intuitive capturing of some important structures in directed graphs. Although overall, the positives outweigh the negatives, leading to the recommendation of acceptance.

**Additional Comments On Reviewer Discussion:**

The reviews were positive from the beginning. The author rebuttal was helpful in clarifying some questions and they added additional experiments, which we encourage them to include in the camera ready. They also agreed to add even further experiments, e.g. on comparisons with "Directed Graph Transformers" which we hope they will include.

---

### Decision · Program_Chairs · 2025-01-22

Accept (Poster)